# Optogenetic manipulation of a value-coding pathway from the primate caudate tail facilitates saccadic gaze shift

Hidetoshi Amita [1,2✉], Hyoung F. Kim[3], Ken-ichi Inoue [2,4], Masahiko Takada[2] & Okihide Hikosaka [1✉]

In the primate basal ganglia, the caudate tail (CDt) encodes the historical values (good or bad) of visual objects (i.e., stable values), and electrical stimulation of CDt evokes saccadic eye movements. However, it is still unknown how output from CDt conveys stable value signals to govern behavior. Here, we apply a pathway-selective optogenetic manipulation to elucidate how such value information modulates saccades. We express channelrhodopsin-2 in CDt delivered by viral vector injections. Selective optical activation of CDt-derived terminals in the substantia nigra pars reticulata (SNr) inhibits SNr neurons. Notably, these SNr neurons show inhibitory responses to good objects. Furthermore, the optical stimulation causes prolonged excitation of visual-saccadic neurons in the superior colliculus (SC), and induces contralateral saccades. These SC neurons respond more strongly to good than to bad objects in the contralateral hemifield. The present results demonstrate that CDt facilitates saccades toward good objects by serial inhibitory pathways through SNr.

[1] Laboratory of Sensorimotor Research, National Eye Institute, National Institutes of Health, Bethesda MD 20892, USA. [2] Systems Neuroscience Section, Primate Research Institute, Kyoto University, Inuyama, Aichi 484-8506, Japan. [3] School of Biological Sciences, Seoul National University, Seoul 08826, Republic of Korea. [4] PRESTO, Japan Science and Technology Agency, Kawaguchi, Saitama 332-0012, Japan. ✉email: amita.hidetoshi@gmail.com; oh@lsr.nei. nih.gov

irecting the eyes toward desirable objects based on past experience is an important skill underlying efficient visual search for food and other high-value items[1,2]. What neuronal network mediates this skill? A network model involving especially the rostral aspect of the basal ganglia explains the mechanism of saccadic eye movements:[3] First, excitation of the caudate nucleus (CD) causes inhibition of the substantia nigra pars reticulata (SNr)[4–6], which then causes excitation of the superior colliculus (SC) via disinhibition[7,8]. Finally, this excitation of SC evokes the contralateral saccade[9–15]. However, no experimental evidence so far has directly shown that excitation of CD leads to disinhibition of SC.

On the other hand, neurons in the caudal aspect of the basal ganglia and SC encode the historical value of many visual objects learned through exposure over long time periods[16–21]. Such stable value coding is reflected by stable differential responses to good objects (previously associated with larger reward) vs. bad objects (previously associated with smaller or no reward) after object-reward association learning[18,22,23]. The circuitry of the caudal aspect of the basal ganglia parallels that of the rostral aspect: the tail of the caudate nucleus (CDt) projects to the caudal–dorsal–lateral SNr (cdlSNr or the substantia nigra pars lateralis)[24], which in turn sends projection fibers to SC[16,25].

Taken together, these studies suggest that the direct pathway from CDt (i.e., the CDt-cdlSNr pathway) controls saccadic eye movements based on reward history through the disinhibition of SC. This hypothesis is further supported by the fact that electrical stimulation of CDt induces contralateral saccades[26]. However, testing the role of the direct pathway in governing behavior has long been complicated by the presence of the indirect pathway, which likewise originates in CDt and then passes through the external globus pallidus (GPe), especially its caudal-ventral part (cvGPe)[19,24]. In order to isolate the direct pathway and elucidate what information CDt transmits to cdlSNr, we employed a pathway-selective optogenetic manipulation that was previously applied to the oculomotor-related pathway from the frontal eye field to SC in primates[27].

To examine how the direct and indirect pathways from CDt encoding the stable value modulate saccadic eye movements, we optogenetically activated either one of the pathways. This was achieved by optical stimulation in either cdlSNr or cvGPe of the axon terminals of CDt neurons. We found that the selective activation of the direct CDt-cdlSNr pathway conveying stable value signals facilitated contralateral saccades through the disinhibition of SC, whereas the activation of the indirect CDt-cvGPe pathway showed no significant effect on saccades. Therefore, we conclude that the multisynaptic CDt-cdlSNr-SC pathway encoding the stable value plays a crucial role in generation of saccades toward good objects within the contralateral hemifield.

## Results

**Opsin expression in CDt neurons**. In order to express channelrhodopsin-2 (ChR2) in the CDt neurons, we injected an adeno-associated virus type 2 vector (AAV2-CMV-ChR2-EYFP)[27] into the CDt, as shown in Fig. 1a. In training sessions carried out many days before the injection, the subjects learned the stable values of many fractal objects that were consistently paired with either a larger reward (good objects) or smaller reward (bad objects) (Fig. 1b). This learning led to automatic gaze bias toward good objects, which persisted for a long time (>1 month) without further learning, in object free-viewing task[16,28].

Prior to each viral vector injection we confirmed that the injectrode was successfully placed in CDt by recording neuronal activity during the passive viewing task (Fig. 1c). Sites within CDt were identified by recording from medium spiny neurons (MSNs,

GABAergic projection neurons), which showed value selective activity (Fig. 1d, $P < 0.001$, Mann–Whitney $U$ test). In each subject we injected the viral vector in four CDt sites spaced at 2 mm intervals along the anterior-posterior axis (light-green dots in Fig. 1e; Top four panels: T1-weighted images of monkey SH; Bottom four panels: T2-weighted images of ZB).

To investigate whether CDt neurons expressed opsin, we recorded neuronal activity in and around CDt and shined blue laser light (473 nm) (Fig. 2a). We identified three groups of CDt neurons based on their spontaneous firing rates, spike waveforms, and autocorrelograms, according to previous studies[29,30] (Supplementary Fig. 1a–e): MSNs (Supplementary Fig. 1c), tonically active neurons (TANs, cholinergic interneurons, Supplementary Fig. 1d), and fast-spiking interneurons (FSIs, presumed parvalbumin-expressing GABAergic interneurons, Supplementary Fig. 1e). Figure 2b shows a representative MSN that was significantly excited by the optical stimulation ($P = 0.0062$, Wilcoxon signed-rank test). Four out of seven recorded MSNs showed significant excitatory responses ($P < 0.05$, Wilcoxon signed-rank test) (the average response in Fig. 2c, Supplementary Fig. 1c, excitation $(+)$). The other three MSNs exhibited no significant response (Supplementary Fig. 1c, excitation $(-)$). TANs ($n = 6/6$) and FSIs ($n = 5/5$) were invariably excited by the optical stimulation (Supplementary Fig. 1d, e). The different proportions of neurons responding to the optical stimulation may be owing to the difference in the transduction efficiency among cell types[31]. In addition, the lack of excitation in some MSNs (Supplementary Fig. 1c, excitation $(-)$) may be caused by a possible indirect effect through excited FSIs and other interneurons in CDt (Fig. 2a)[32]. These results suggested that the vector was transfected in the three groups of CDt neurons, as shown in Fig. 2a.

Histological results also confirmed the expression of ChR2-EYFP in both monkeys. Representative sections obtained in monkey SH are depicted in Fig. 2d–f. Many ChR2-positive neurons were found in CDt and the putamen tail (PUTt) (Fig. 2d), both of which include stable value-coding neurons[17,18,21]. We also found many ChR2-positive axon terminals in cdlSNr (Fig. 2e) and cvGPe (Fig. 2f). The locations of these ChR2-positive terminals were quite similar to the locations of anterogradely labeled axon terminals of CDt neurons in our previous study[24]. These results together establish that the opsin was transported to the terminals of CDt neurons, which were located in both cdlSNr and cvGPe. Next, we investigated whether pathway-selective optical activation (CDt-cdlSNr or CDt-cvGPe pathway) modulated neuronal activity in the downstream target areas (i.e., cdlSNr or cvGPe).

**Direct pathway conveys stable value signals from CDt**. To investigate the properties of SNr neurons receiving direct inputs from CDt, we shined light on the axon terminals of CDt and recorded neuronal activity in SNr using an optrode, consisting of a microelectrode and a fiber-optic cable (Fig. 3a). Figures 3b and c show a representative neuron in cdlSNr. This neuron was inhibited completely by brief optical stimulation (20 ms, 955 mW mm$^{-2}$) (Fig. 3b, $P < 0.001$, Wilcoxon signed-rank test). This response pattern was similar to the effect of orthodromic electrical stimulation of CDt[22]. In recordings taken during the passive viewing task (Fig. 1c), the same neuron was inhibited by four good objects and excited by four bad objects (Fig. 3c, $P < 0.001$, Mann–Whitney $U$ test), indicating that the neuron encoded stable values of visual objects.

Among 109 SNr neurons, 66 neurons were inhibited by the optical stimulation of axon terminals within cdlSNr and none were excited (inhibition $(+)$; Fig. 3d), confirming that the output

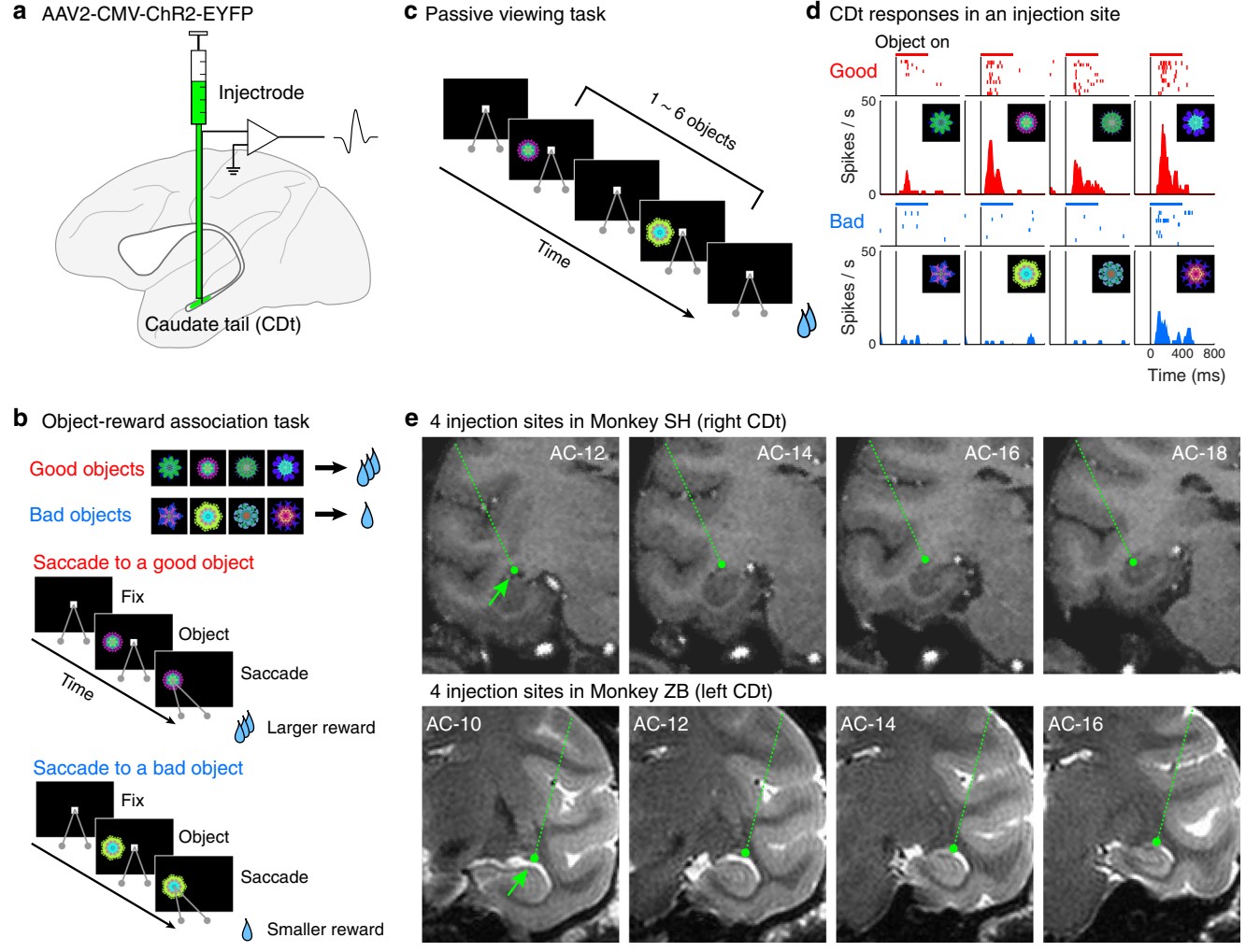

**Fig. 1 Injection of ChR2 vector in stable value-coding CDt. a** Viral vector was injected into CDt using an injectrode. Injection sites were guided by single-unit recording. **b** Object-reward association task. Eight fractal objects in a set were used in each session, four of which were associated with a larger reward (good objects) and four with a smaller reward (bad objects). Subjects made a saccade to a peripheral good or bad object after a center fixation dot disappeared. After repeated learning, the subject chose good objects automatically in object free-viewing task[20]. **c** Passive viewing task. The same eight objects previously used in the object-reward association task were sequentially presented in a random order. Subjects were rewarded after fixating a center dot until its disappearance irrespective of what objects appeared. **d** A representative neuron in an injection site in CDt showing stable object value signals. This neuron showed significantly higher response to good objects (top) than bad objects (bottom) ($P = 9.5 \times 10^{-6}$, Mann–Whitney U test, two-sided). Each raster plot indicates spike firing timings aligned to the presentation onset of each object. Each peri-stimulus time histogram (PSTH) indicates responses to each object. **e** Estimated injection sites on coronal planes of magnetic resonance images. Green dots indicate four injection sites in CDt for each subject. Green dotted lines show approaches to the injection sites. Top four panels: T1-weighted images of monkey SH; bottom four panels: T2-weighted images of ZB. Labels indicate slices 10, 12, 14, 16, and 18 mm posterior to the anterior commissure (AC).

of the striatum is inhibitory[5,6,33]. The average latency of inhibitory response was 5.8 ms (Supplementary Fig. 2). This timing is shorter than the average latency of spikes in cdlSNr neurons evoked by orthodromic electrical stimulation in CDt (8.8 ms)[22], suggesting that the optical stimulation directly activated the axon terminals of CDt as expected. Note that the brief inhibitory responses to the optical stimulation were also observed even when the stimulation duration was 100 ms (Supplementary Fig. 3a). We then found that the optically inhibited neurons, as a population, encoded stable values (Fig. 3e, $P < 0.001$, Wilcoxon signed-rank test): An early excitation (visual response latency: ~50 ms) was followed by an inhibition by good objects or a further excitation by bad objects (value-coding latency: ~100 ms). Among individual neurons, many (28 among 66) showed significant negative value-coding ($P < 0.05$, negative value in Fig. 3f; lower responses to good objects than bad objects).

These negative value-coding neurons, as a population, displayed significant inhibitory responses to good objects ($P = 0.026$, Wilcoxon signed-rank test) and significant excitatory responses to bad objects ($P < 0.001$, Wilcoxon signed-rank test) (Supplementary Fig. 4a), as previously demonstrated[16,34]. In contrast, the other neurons (38 among 66) showed excitatory responses to both good and bad objects with short latencies (Supplementary Fig. 4b), suggesting that they were likely to receive input not only from CDt, but also from other regions. These SNr neurons, which received inhibitory input from CDt, were located in the cdlSNr area (cyan circles in Fig. 3j).

Another group of SNr neurons (43 among 109) showed no significant responses to optical stimulation (Inhibition (−); Fig. 3g), suggesting that they may not receive inhibitory input from CDt. During the passive viewing task, the population activity of this Inhibition (−) group started with an early

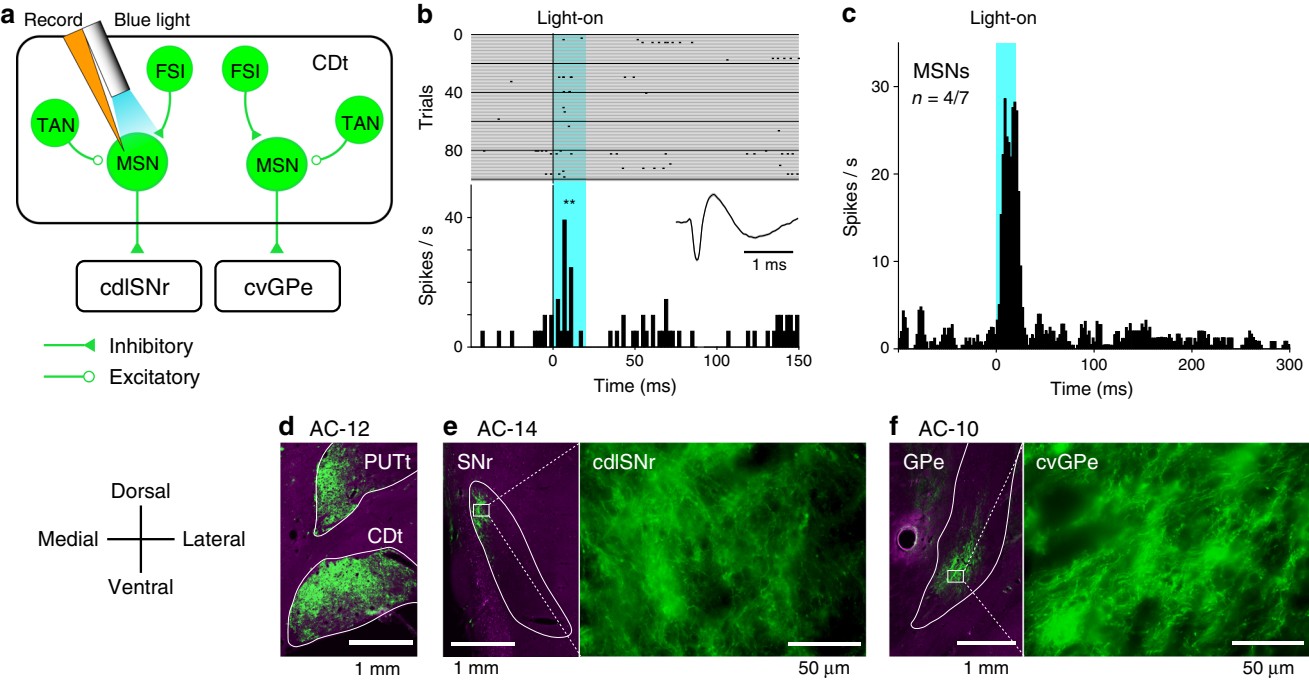

**Fig. 2 Optically evoked responses and ChR2-expressing CDt axon terminals in SNr and GPe. a** Recording from CDt neurons during optical stimulation in CDt. MSNs in CDt receive inhibitory inputs from FSIs and excitatory inputs from TANs, and have projections to cdlSNr or cvGPe. The ChR2 vector was non-selectively transfected in all groups of CDt neurons. **b** A representative MSN showing an excitatory response to optical stimulation (20 ms duration) ($P = 0.0062$, Wilcoxon singed-rank test, two-sided). Raster plots show the activity aligned to the stimulation onset. PSTHs show the average response of the neuron to the stimulation across trials. Light-blue shade indicates the stimulation period (0–20 ms). An inset shows the MSN's spike waveform. **indicates $P < 0.01$. **c** The average activity of four out of seven MSNs showing significant excitatory responses to the optical stimulation ($P < 0.05$, Wilcoxon singed-rank test, two-sided). **d** ChR2-positive neurons in CDt and PUTt of monkey SH. Green signal indicates enhanced EYFP signal. Scale bar: 1 mm. Histological results of monkey ZB replicated the ChR2-positive neurons in CDt and PUTt. **e** Left: ChR2-positive axon terminals in cdlSNr of monkey SH. Scale bar: 1 mm. Right: enlarged image of the axons in cdlSNr (the area shown by white box in left). Scale bar: 50 μm. The results of monkey ZB replicated the ChR2-positive axon terminals in cdlSNr. **f** Left: ChR2-positive axon terminals in cvGPe of monkey SH. Right: enlarged image of the axons in cvGPe (the area shown by white box in left). Scale bar: 50 μm. The results of monkey ZB replicated the ChR2-positive axon terminals in cvGPe.

excitation (latency: ~50 ms in Fig. 3h, similar to the Inhibition (+) group (Fig. 3e), but was not followed by differential responses to good and bad objects (Fig. 3h, $P = 0.77$, Wilcoxon signed-rank test). It should also be noted that their early excitatory responses to good and bad objects were similar to the early responses of 38 neurons in the Inhibition (+) group not representing significant negative values (Supplementary Fig. 4b). As individual neurons, only six neurons encoded stable values (Fig. 3i). These SNr neurons were located in cdlSNr as well as other parts of SNr (black circles in Fig. 3j).

These results suggest that the inhibitory responses of cdlSNr neurons to good objects are caused by the direct inhibitory input from CDt. However, the excitatory responses to bad objects are unlikely caused by this direct input (Fig. 3e, Supplementary Fig. 4a), considering that the baseline activity of MSNs in CDt is low frequency[16,24,27] and MSNs are inhibitory[5,6]. To explore this issue further, we turned to cvGPe, which is another projection site of CDt (Fig. 2f) and has inhibitory neuron responses to bad objects[19,24].

**Indirect pathway of CDt transfers opposite stable values.** To investigate the activity of GPe neurons receiving monosynaptic inputs from CDt, we recorded neuronal activity in GPe and optically stimulated the axon terminals from CDt using the optrode (Fig. 4a). Figure 4b–c shows a representative cvGPe neuron. This neuron was inhibited briefly followed by excitation by a brief optical stimulation (20 ms, 318 mW mm$^{-2}$) (Fig. 4b, $P < 0.001$,

Wilcoxon signed-rank test). During the passive viewing task (Fig. 1c), the neuron was inhibited by four bad objects (Fig. 4c, $P = 0.0079$, Mann–Whitney $U$ test). This indicates that the cvGPe neuron encoded stable values of visual objects, similarly to the cdlSNr neurons (Fig. 3c) but in the opposite manner.

Among 78 GPe neurons, 61 neurons were initially inhibited by the optical stimulation, often followed by an excitation (Inhibition (+); Fig. 4d). The average latency of the inhibitory responses was 4.6 ms (Supplementary Fig. 2). These optically evoked latency and rebound response patterns are similar to the responses of cvGPe evoked by orthodromic electrical stimulation of CDt[19]. Note that the brief inhibitory responses to the optical stimulation were also observed even when the stimulation duration was 100 ms (Supplementary Fig. 3b). We then found that the optically inhibited neurons, as a population, encoded stable values (Fig. 4e, $P = 0.0067$, Wilcoxon signed-rank test). An early excitation (latency: ~50 ms) was followed by an inhibition by bad objects or a further excitation by good objects (value-coding latency: ~100 ms). Among individual neurons, many of them (22 among 61) showed significant positive value-coding ($P < 0.05$, Positive value in Fig. 4f; lower responses to bad objects than good objects). These positive value-coding neurons displayed significant inhibitory responses to bad objects ($P = 0.0024$, Wilcoxon signed-rank test), but no significant responses to good objects ($P = 0.41$, Wilcoxon signed-rank test) (Supplementary Fig. 4c), as previously reported[19,34]. By contrast, the other neurons (39 among 61) exhibited excitatory responses to both good and bad objects with short latencies (Supplementary Fig. 4d), suggesting that they

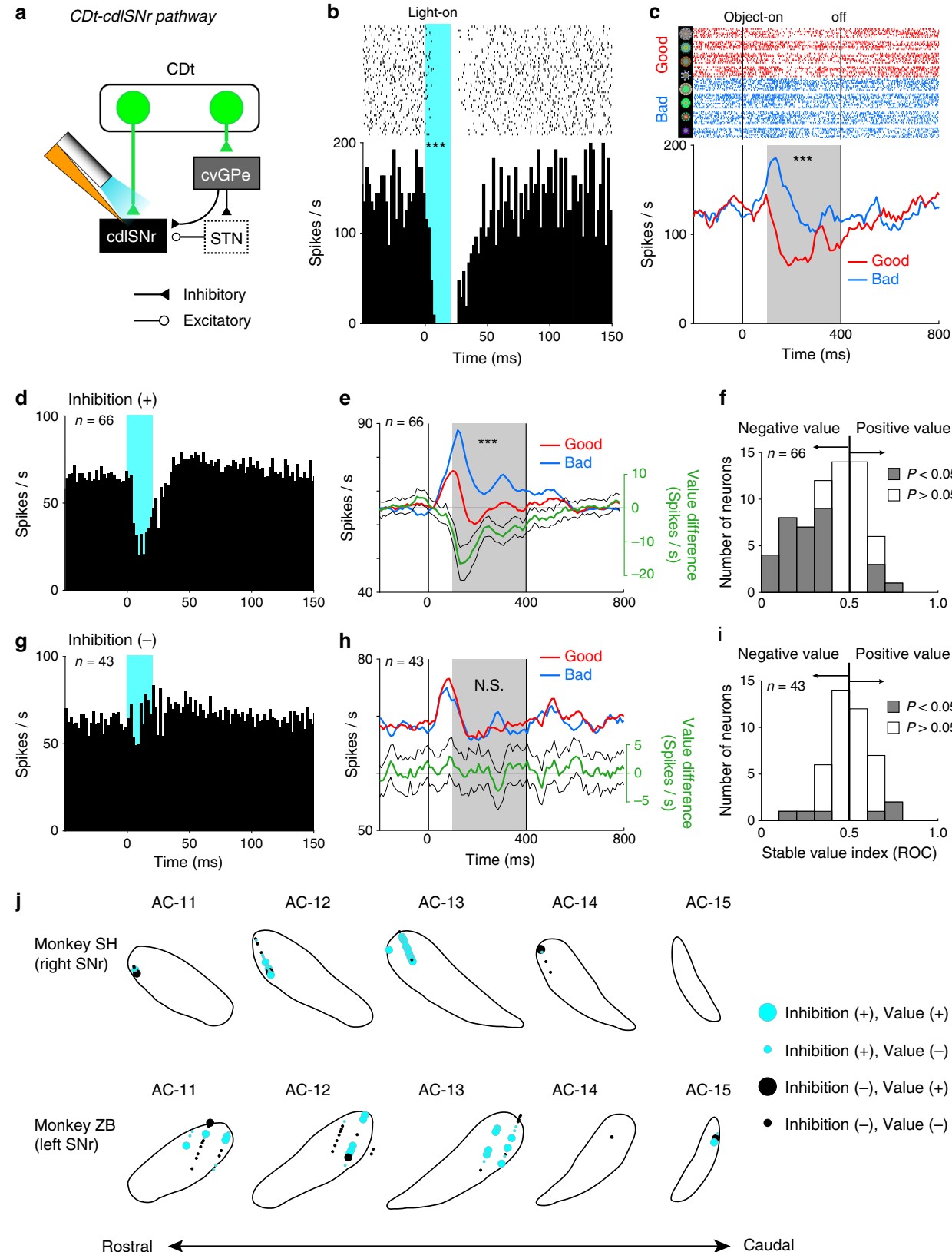

were likely to receive input not only from CDt, but also from other regions. These GPe neurons, which received inhibitory input from CDt, were all located in cvGPe (cyan circles in Fig. 4j).

Another group of GPe neurons (17 among 78) showed no significant responses to optical stimulation (Inhibition (−); Fig. 4g). During the passive viewing task, their population activity

started with an early excitation (latency: ~ 50 ms), but was not followed by differential responses to good and bad objects (Fig. 4h, P = 0.52, Wilcoxon signed-rank test). As individual neurons, only three neurons encoded stable values (Fig. 4i). The 17 Inhibition (−) neurons were located both in cvGPe (black circles in Fig. 4j) and also outside cvGPe.

**Fig. 3 Inhibition of stable value-coding SNr neurons by optical stimulation. a** Recording from cdlSNr during optical stimulation to CDt-cdlSNr pathway. **b** A representative cdlSNr neuron showed an inhibitory response to the stimulation ($P = 3.3 \times 10^{-10}$, Wilcoxon signed-rank test, two-sided). Raster and PSTH show the activity aligned on stimulation onset. Light-blue shade indicates the stimulation period. *** indicates $P < 0.001$. **c** The same neuron showed significantly lower responses to good objects than bad in the passive viewing task ($P = 3.1 \times 10^{-7}$, Mann–Whitney $U$ test, two-sided). Raster and PSTH show the activity aligned to the onset of good (red) or bad objects (blue). Light-gray shade shows the statistical test window. *** indicates $P < 0.001$. **d** Average activity of 66 cdlSNr neurons showing significant responses to the stimulation. **e** Population activity of the 66 neurons showed significantly lower responses to good objects than bad ($P = 1.1 \times 10^{-4}$, Wilcoxon signed-rank test, two-sided). Green PSTH shows the value difference (i.e., good−bad) (mean ± SEM). *** indicates $P < 0.001$. **f** Distribution of the 66 neurons across stable value index. Gray and white histograms indicate significant value-coding ($P < 0.05$, Wilcoxon signed-rank test, two-sided) and non-significant neurons ($P > 0.05$, Wilcoxon signed-rank test, two-sided), respectively. **g** Average activity of 43 neurons not showing significant responses to the stimulation. **h** Population activity of the 43 neurons showed no significant difference between good and bad objects ($P = 0.77$, Wilcoxon signed-rank test, two-sided). N.S. indicates $P > 0.05$. **i** Distribution of the 43 neurons across the stable value index. Gray and white histograms indicate significant value-coding ($P < 0.05$, Wilcoxon signed-rank test, two-sided) and non-significant neurons ($P > 0.05$, Wilcoxon signed-rank test, two-sided), respectively. **j** Localization of stimulation-responsive neurons in cdlSNr. Inhibition (+) and (−) indicate neurons showing significant inhibitory responses to the stimulation (cyan) and non-significant responses (black), respectively. Value (+) and (−) indicate significant value-coding neurons (large circles) and non-significant neurons (small circles). Top: monkey SH. Bottom: monkey ZB.

**Disinhibition of SC by activation of CDt-SNr pathway.** Our previous studies suggested that the main ultimate target of CDt is SC[16,19,22,24], but there has been no direct evidence. To test this hypothesis physiologically we conducted simultaneous recordings in both cdlSNr (by optrode) and SC (by recording electrode) (Fig. 5a). A group of SNr neurons projects to SC, mainly to the intermediate layer of SC[35–37] where neurons are active immediately before saccadic eye movements and sometimes in response to visual stimuli[9–12,14,36]. We therefore placed the recording electrode in the intermediate layer of SC when the optical stimulation activated the axon terminals of CDt neurons (which are all inhibitory neurons) on cdlSNr neurons.

In this configuration, we first investigated the responses of a cdlSNr neuron and a SC neuron to the optical stimulation in cdlSNr (Fig. 5a). We let the subject freely watched movies (showing behaving animals) so that the activity of SC neurons was higher and was likely to be affected by a change in inputs. The cdlSNr neuron was inhibited briefly (Fig. 5b), as shown before (Fig. 3d). In contrast, the SC neuron was excited, but apparently for a longer duration (Fig. 5c). The activity was significantly higher in the 100 ms after the stimulation started than the baseline activity ($P = 1.0 \times 10^{-4}$, Wilcoxon signed-rank test). We tested a total 16 SC neurons with optical stimulation of cdlSNr and found that these neurons were significantly excited (Fig. 5e, $P = 0.001$, Wilcoxon signed-rank test). Notably, the excitatory response of SC neurons (Fig. 5e), on average, started later and lasted longer than the inhibitory response of cdlSNr neurons (Fig. 5d, Supplementary Fig. 5a). The cumulative plot of the SC activity also showed prolonged effect after the brief stimulation (Supplementary Fig. 5b).

These results thus far establish that the selective optical stimulation of the CDt-cdlSNr pathway is sufficient to activate SC neurons. This effect was dissociated from the effect of the optical stimulation of the indirect pathway (i.e., CDt-cvGPe pathway), which showed no significant effect on SC neurons (Supplementary Fig. 6b, d).

To investigate whether the cdlSNr-SC connection coveys value signals, we recorded the activity of cdlSNr and SC neurons simultaneously (as shown in Fig. 5a) during the object-reward association task (Fig. 5f). In each trial, one good or bad object was presented at the location of the SC neuron's visual receptive field and the subject was required to make a saccade to the object after the fixation dot disappeared. Figure 5h shows the population activity of five pairs of simultaneously recorded cdlSNr and SC neurons. Before starting this task, all neuron pairs showed significant responses to the optical stimulation in cdlSNr, similarly to the data in Fig. 5d–e. The cdlSNr neurons were inhibited by good objects more strongly than bad objects (Fig. 5h,

middle-left, $F(4, 4) = 0.57$, $P = 0.60$, $F$ test; $P = 0.0045$, paired $t$ test), whereas the SC neurons were excited by good objects more strongly than bad objects (Fig. 5h, bottom-left, $F(4, 4) = 1.46$, $P = 0.72$, $F$ test; $P = 0.0083$, paired $t$ test). Moreover, the subject made significantly earlier saccades toward good objects than toward bad objects after the fixation dot disappeared (Fig. 5g, $P = 0.0079$, Mann–Whitney $U$ test). To investigate whether the cdlSNr and SC neurons may contribute to the earlier saccades toward good objects, we analyzed their firing rates in a window 0–100 ms after fixation offset, which included the mean latency of saccades toward good objects. The cdlSNr activity remained more inhibited by good objects than bad objects during this period (Fig. 5h, middle-right, $F(4, 4) = 0.72$, $P = 0.75$, $F$ test; $P = 0.021$, paired $t$ test). Correspondingly, the SC activity was more excited by good objects than bad objects in the same period (Fig. 5h, bottom-right, $F(4, 4) = 0.97$, $P = 0.98$, $F$ test; $P = 0.017$, paired $t$ test).

These results suggest that the shorter latency saccades to good objects occurred because disinhibition within the CDt-cdlSNr-SC network was stronger in response to good objects than to bad objects. Thus, the generation of the saccadic signal by SC neurons would be modulated the value signal. If this interpretation is correct, then excitation of these visual-saccadic SC neurons by optical stimulation of the CDt-cdlSNr pathway should induce saccadic eye movements. This hypothesis was tested by optical stimulation, as described below.

**Facilitation of contralateral saccade via CDt-SNr-SC network.** We have so far shown that optical stimulation of the pathway originating in CDt modulated downstream neurons (cdlSNr, cvGPe, and SC). Here we ask a final question: does the same optical stimulation modulate eye movements? To address this question, we let the subject freely watch movies, whereas cdlSNr was optically stimulated occasionally (Fig. 6a: movie free-viewing task). Two considerations dictated the use of movies for this experiment. First, neurons in CDt, cdlSNr, and cvGPe are all sensitive to and selective for complex visual stimuli[16,19,26,38]. Second, the prevalence of spontaneous saccades during free viewing afforded the opportunity to detect changes in multiple saccade parameters, including latency, direction, duration, and frequency of saccades.

Figure 6b (top) shows the spike activity of an SC neuron (same as in Fig. 5c) and saccades (orientation and amplitude) before and after each optical stimulation in cdlSNr. After optical stimulation in cdlSNr of the axon terminals of CDt neurons (Fig. 6a), the SC neuron increased its firing rate (Fig. 6b, bottom), often with a burst of spikes (Fig. 6b, top), and

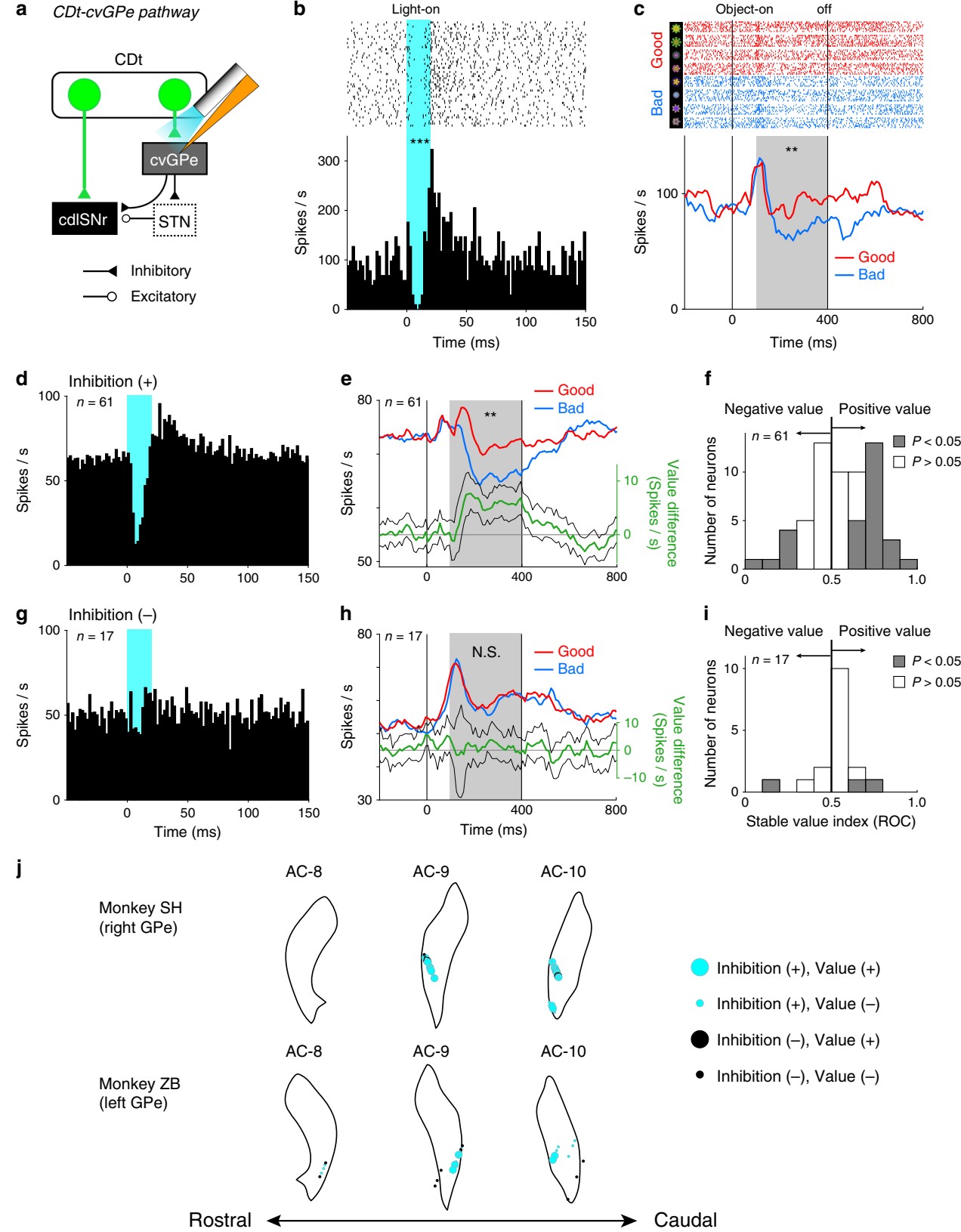

saccades were directed to the contralateral side more often than before the stimulation (Fig. 6b, c). Spike bursts were sometimes followed by relatively small saccades (amplitude <20°) to the contralateral side (red arrows in Fig. 6b) which were close to the visual receptive field of the SC neuron (Fig. 6c, right).

To investigate the effect of optical stimulation in cdlSNr on eye movements across all sessions, we plotted the cumulative difference in the number of contralateral saccades and ipsilateral saccades (Fig. 6d). The results indicate that the prevalence of contralateral saccades continued for ~200 ms both monkeys (Fig. 6d: monkey SH, Fig. 6e: monkey ZB), even though the

**Fig. 4 Inhibition of stable value-coding GPe neurons by optical stimulation. a** Recording from cvGPe during optical stimulation of CDt-cvGPe pathway. **b** A representative cvGPe neuron showed an inhibitory response to the stimulation ($P = 9.9 \times 10^{-8}$, Wilcoxon signed-rank test, two-sided). *** indicates $P < 0.001$. Same format as Fig. 3b. **c** The same neuron showed significantly lower responses to bad objects than good in the passive viewing task ($P = 0.0079$, Mann–Whitney $U$ test, two-sided). ** indicates $P < 0.01$. Same format as Fig. 3c. **d** Average activity of 61 cvGPe neurons showing significant responses to the stimulation. Same format as Fig. 3d. **e** Population activity of the 61 neurons showed significantly higher response to good objects than bad ($P = 0.0067$, Wilcoxon signed-rank test, two-sided). ** indicates $P < 0.01$. Same format as Fig. 3e. **f** Distribution of the 61 neurons across stable value index. Gray and white histograms indicate significant value-coding ($P < 0.05$, Wilcoxon signed-rank test, two-sided) and non-significant neurons ($P > 0.05$, Wilcoxon signed-rank test, two-sided), respectively. **g** Average activity of 17 neurons that did not show significant responses to the stimulation. **h** Population activity of the 17 neurons to objects showed no significant difference between good objects and bad objects ($P = 0.52$, Wilcoxon signed-rank test, two-sided). N.S. indicates $P > 0.05$. **i** Distribution of the 17 neurons across stable value index. Gray and white histograms indicate significant value-coding ($P < 0.05$, Wilcoxon signed-rank test, two-sided) and non-significant neurons ($P > 0.05$, Wilcoxon signed-rank test, two-sided), respectively. **j** Localization of stimulation-responsive neurons in cvGPe. Same format as Fig. 3j.

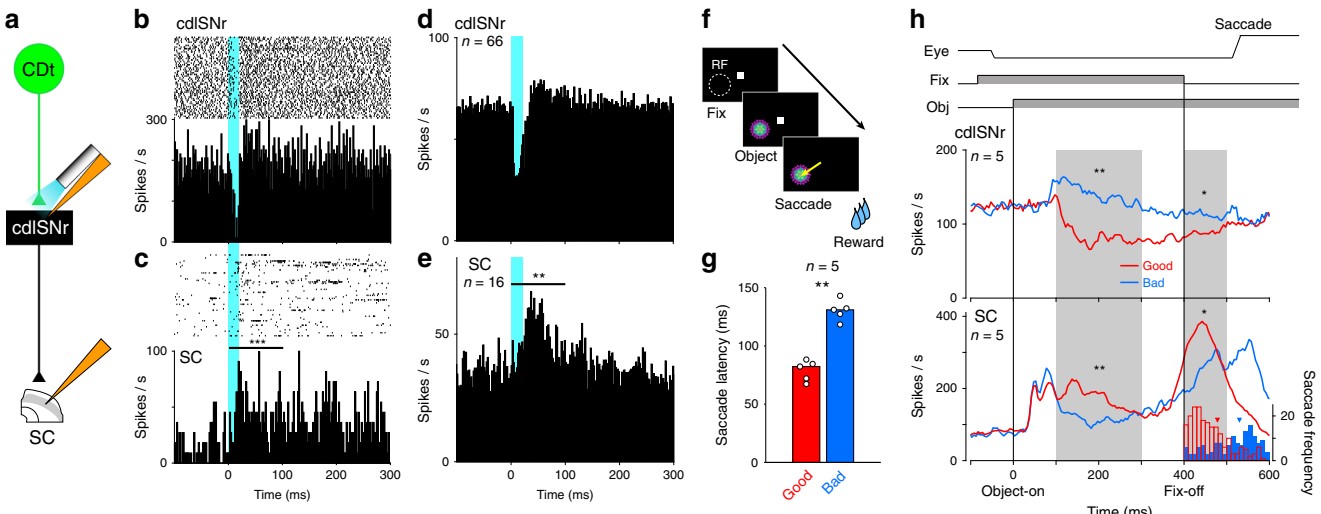

**Fig. 5 Disinhibition of value-coding neuron in SC by optical stimulation to CDt-SNr pathway. a** Simultaneous recordings of neuronal activity in cdlSNr and SC during optical stimulation of CDt-cdlSNr pathway. **b** Responses of a representative cdlSNr neuron to the stimulation. **c** Responses of a representative SC neuron to the stimulation ($P = 1.0 \times 10^{-4}$, Wilcoxon signed-rank test, two-sided). *** indicates $P < 0.001$. **d** Averaged responses of 66 cdlSNr neurons to the stimulation (same data as Fig. 3d). **e** The population activity of 16 SC neurons showed significantly more excitation in post-stimulation than pre-stimulation ($P = 0.001$, Wilcoxon signed-rank test, two-sided). ** indicates $P < 0.01$. **f** Object-reward association task. In each trial, either good or bad object was presented at the SC neuron's receptive field (white dotted circle). **g** Saccade latency to good objects was significantly shorter than bad in the task ($n = 5$ sessions; $P = 0.0079$, Mann–Whitney $U$ test, two-sided). Each bar indicates median of saccade latency to good (red) or bad objects (blue). Each filled circle depicts mean latency in a session. ** indicates $P < 0.01$. **h** The population activity of cdlSNr and SC neurons during the task. Top: The time schedule. *Eye* eye movement, *Fix* fixation dot presentation, *Obj* object presentation. Middle: inhibitory response of five cdlSNr neurons to good objects (red) showed significantly stronger than bad (blue) after object onset ($P = 0.0045$, paired $t$ test, two-sided) and fixation offset ($P = 0.021$, paired $t$ test, two-sided). Bottom: excitatory response of five SC neurons to good objects (red) was significantly stronger than bad (blue) after object onset ($P = 0.0083$, paired $t$ test, two-sided) and fixation offset ($P = 0.017$, paired $t$ test, two-sided). Histograms and inverted triangles indicate the distributions of saccade latencies and the average latencies to good (red) and bad objects (blue). * and ** indicate $P < 0.05$ and 0.01, respectively.

optical stimulation lasted only 20 ms. This prolonged saccade bias corresponded to the duration of elevated SC responses to the optical stimulation (Supplementary Fig. 5b). Across all trials, the mean number of contralateral saccades increased, whereas the numbers of ipsilateral saccades decreased within 200 ms after stimulation onset (Fig. 6f, g). The impact of optical stimulation on mean saccade magnitude and direction is shown separately for monkeys SH (Fig. 6h) and ZB (Fig. 6i). Similar direction biases occurred for the first saccade after the stimulation (Supplementary Fig. 7a–d). Optical stimulation in cvGPe did not cause a significant change in saccade direction (Supplementary Fig. 8a–f).

Our results provide evidence that value signals associated with good objects are conveyed selectively through the direct CDt-cdlSNr-SC network, and consequently saccades to high-value objects are facilitated in visual search.

## Discussion

In the present study, the following two technical approaches were combined in the primate brain: electrophysiological recording of task-related neuronal activity and its optogenetic manipulation in a pathway-selective fashion. The former approach has revealed that numbers of CDt, cdlSNr, and SC neurons represent the stable value of good objects, as previously reported[16–18,20–22]. The latter approach, on the other hand, has revealed that the CDt-cdlSNr-SC network initiates saccadic eye movements by disinhibition via the two sequential inhibitory pathways (i.e., the CDt-cdlSNr and cdlSNr-SC pathways). In our experimental paradigm, the same neuronal populations sampled by single-unit recordings were modulated by optical stimulation. Therefore, we conclude that stable value signals transmitted along the CDt-cdlSNr-SC network subserve saccadic gaze shifts toward good objects within the contralateral hemifield.

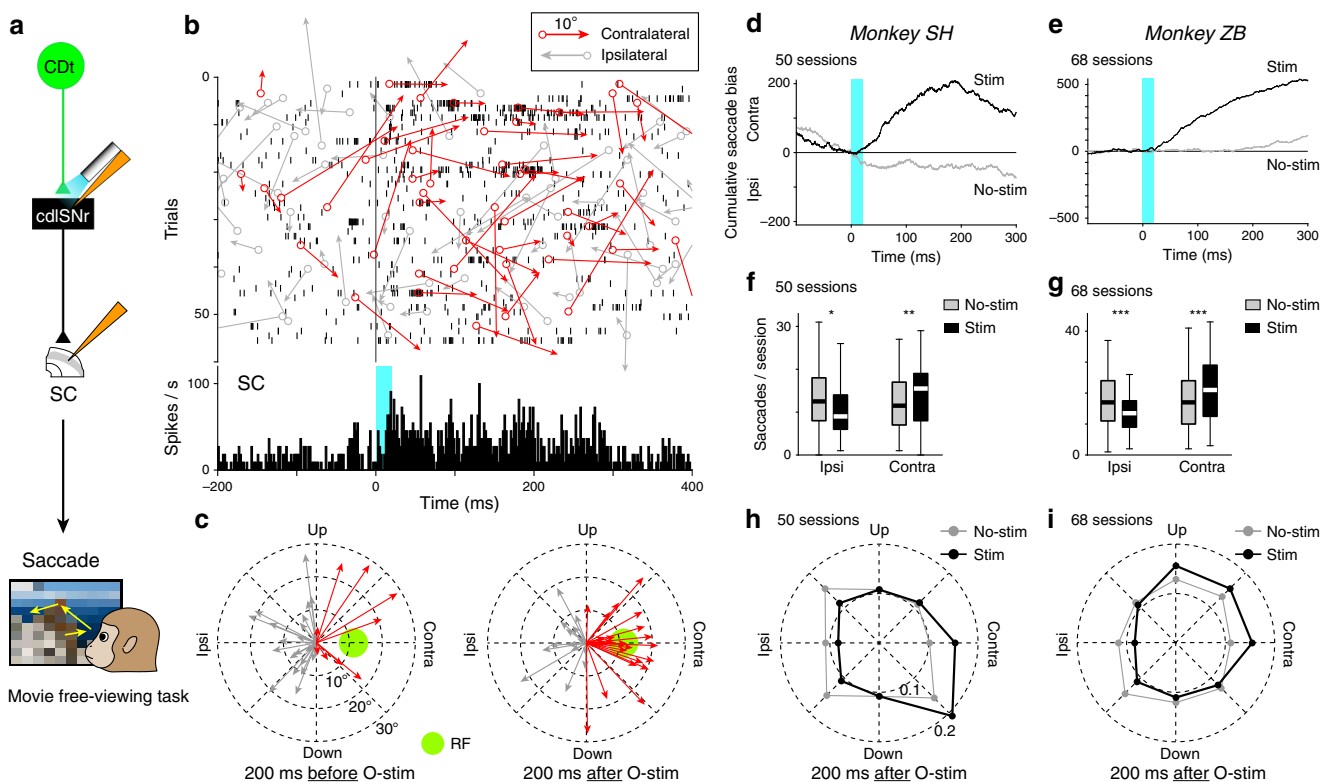

**Fig. 6 Facilitation of contralateral saccade by optical stimulation to CDt-SNr pathway. a** Eye movements before and after optical stimulation were examined in the movie free-viewing task. **b** Effects of optical stimulation on SC neuronal activity and saccades. Activity of an SC neuron is shown by raster (top) and spike density function (bottom) (same data as Fig. 5c). Red and gray arrows indicate contralateral and ipsilateral saccades. Angle and length of arrows show direction and amplitude. Each open circle shows the saccade onset along the raster. **c** Saccade directions in 200 ms before (left) vs. after (right) the stimulation (same data in Fig. 6b). Red and gray arrows show contralateral and ipsilateral saccades, respectively. Light-green circle shows the SC neuron's visual receptive field. **d** Temporal dynamics of saccade bias in monkey SH (total 50 sessions). Black and gray curves indicate the cumulative saccade bias in stimulation trials (Stim) vs. no-stimulation trials (No-stim). Light-blue shade indicates the stimulation period. **e** Temporal dynamics of saccade bias in monkey ZB (total 68 sessions). **f** Number of saccades per session in 200 ms after no-stimulation (gray) vs. stimulation (black) in monkey SH. Ipsilateral saccades per session after the stimulation significantly decreased ($n = 50$, $P = 0.024$, Wilcoxon singed-rank test, two-sided), whereas contralateral saccades significantly increased ($n = 50$, $P = 0.0015$, Wilcoxon singed-rank test, two-sided). The band indicates the median, the box indicates the first and third quartiles and the whiskers indicate ± 1.5 × interquartile range. *, ** indicate $P < 0.05$, 0.01. **g** In monkey ZB, ipsilateral saccades after the stimulation significantly decreased ($n = 68$, $P = 8.2 \times 10^{-7}$, Wilcoxon singed-rank test, two-sided), whereas contralateral saccades significantly increased ($n = 68$, $P = 5.9 \times 10^{-5}$, Wilcoxon singed-rank test, two-sided). Same format as Fig. 6f. *** indicates $P < 0.001$. **h** Mean saccade directions within 200 ms after no-stimulation (gray) vs. stimulation (black) in monkey SH. **i** Mean saccade directions in monkey ZB.

A series of previous studies have shown that neurons in each structure constituting the CDt-cdlSNr-SC network encode stable values of visual objects in the passive viewing task[16–18,20–22]. However, it remains to be solved how these neurons convey and transfer the value signals to induce saccadic eye movements. Initially, CDt neurons receive visual object information mainly from the temporal cortex[39–42]. This information can be transferred to stable value signals (mostly good value signals) through the CDt-cdlSNr pathway (see Fig. 7). This scheme is supported by the present results that the same cdlSNr neurons that are inhibited by optical stimulation also show inhibitory responses to good objects (see Fig. 3d-f, Supplementary Fig. 4a). The inhibition of cdlSNr would lead to activation of SC neurons[35] (see Fig. 7).

Unpredictably, a brief stimulation (20 ms) of the CDt-cdlSNr pathway caused a prolonged facilitation (~200 ms) of SC neuron activity (see Supplementary Fig. 5a–b) and evoked contralateral saccades (see Fig. 6d-e and Supplementary Fig. 5c). This event may be ascribed to mutual excitatory connections between SC neurons (see Fig. 7) which have been shown both anatomically and physiologically[43–45]. Such connectivity appears to facilitate production of burst discharges that drive the SC network towards

threshold for saccade generation[9–14]. Once a phasic excitation (disinhibition) of SC is triggered by the inhibition of cdlSNr, SC neurons may be activated continuously through their mutual connections until saccades occur. It is most likely that this neuronal mechanism enables the visual-saccadic SC neurons to transfer the value signal to the saccade signal (see Fig. 5h).

Of particular importance is that the CDt-cdlSNr-SC network conveys the spatial information about good objects. As previously reported, individual neurons in this network have receptive fields at the contralateral periphery[16–18,20,26]. It has been found that an SNr neuron projects to a specific site of SC where neurons have visual or oculomotor fields similar to the SNr neuron[35]. In favor of this, our data showed that pairs of cdlSNr and SC neurons, which responded to the optical stimulation of the CDt-cdlSNr pathway, shared their receptive fields (see Fig. 5h). Accordingly, the information concerning a good object presented at a given position might be conveyed topographically along the CDt-cdlSNr-SC network in which individual neurons have similar receptive fields. Thus, saccadic gaze shifts toward good objects within the contralateral hemifield could successfully be achieved even in the case where many bad objects exist around the good object.

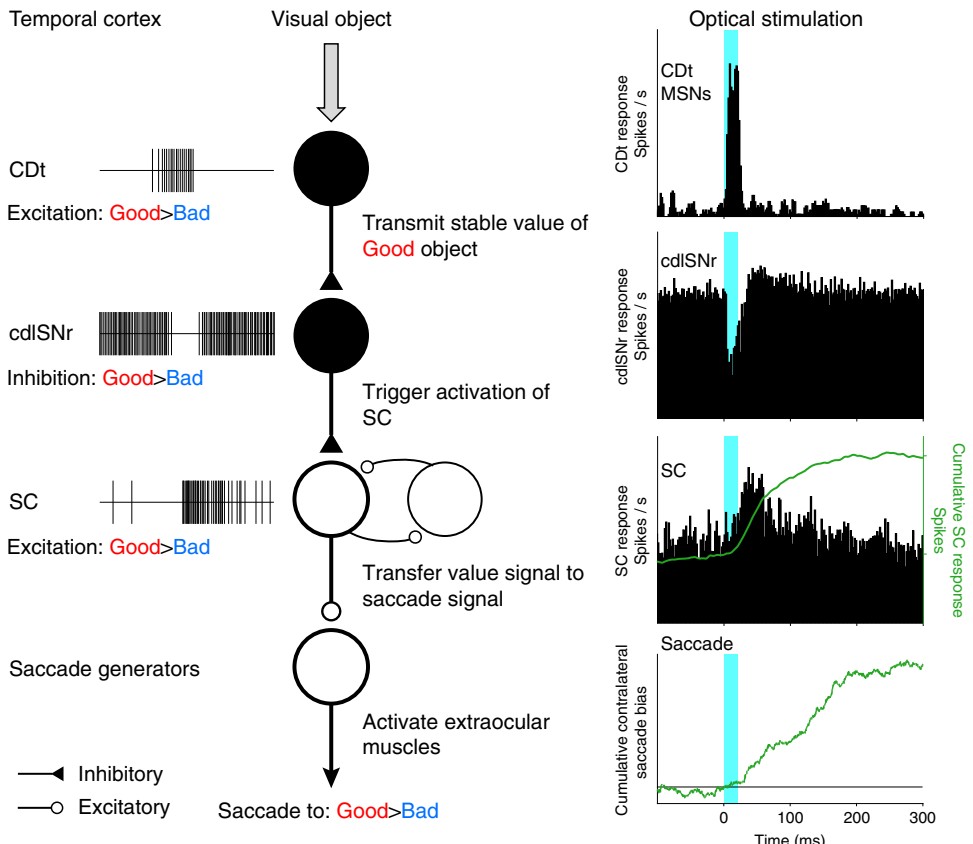

**Fig. 7 Schematic model of CDt-SNr-SC network for gaze shift toward good objects.** Left: CDt receives visual object information mainly from the temporal cortex. The CDt creates and transmits stable value signals indicating good objects to cdlSNr. The cdlSNr is inhibited by the input from CDt, and triggers activation of SC. In SC, the value signal is transferred to saccade signal. Finally, the signal reaches to saccade generators for generating a saccade to the good object. Right: optogenetic manipulation imitates the value signal processing of good objects thorough the CDt-cdlSNr-SC network (left). CDt neurons are excited by the optical stimulation, cdlSNr neurons are inhibited, then SC neurons are disinhibited. Cumulative plot of SC response to the stimulation (green line in the third panel from the top, see Supplementary Fig. 5b) corresponds to the cumulative plot of contralateral saccade bias by the stimulation (the fourth panel from the top, see Supplementary Fig. 5c).

We have demonstrated that the stable value signals synthesized in CDt are transmitted not only to cdlSNr (see Fig. 3), but also to cvGPe (see Fig. 4). The cdlSNr neurons were more frequently inhibited by good than bad objects (see Fig. 3e, Supplementary Fig. 4a), and vice versa for the cvGPe neurons (see Fig. 4e, Supplementary Fig. 4c). This finding suggests that the indirect pathways through cvGPe (i.e., the CDt-cvGPe-cdlSNr and, also, CDt-cvGPe-subthalamic nucleus (STN)-cdlSNr pathways; see Figs. 3a and 4a) convey stable value signals in a manner opposite to that of the direct CDt-cdlSNr pathway. Our recent work has shown that pharmacological blockade of cvGPe neuron responses to bad objects results in the attenuation of cdlSNr neuron activity enhanced by bad objects[34]. Accordingly, we predicted that the optical stimulation of the CDt-cvGPe pathway could lead to the inhibition of SC neuron activation and consequent saccades, however, this effect was not observed in the present study (see Supplementary Fig. 6b, d; Supplementary Fig. 8a–f). This might be ascribed to the complex pattern of heterogeneous GPe neuron connectivity[46–52]. In addition to the indirect pathways described above (CDt-cvGPe-(STN)-cdlSNr pathways), cvGPe neurons may have reciprocal connections with STN neurons[51,52] or striatal neurons[46], or have mutual inhibitory connections within GPe[50]. Thus, the optical stimulation might reduce the presumed effect on the indirect pathways.

Anatomical studies have reported that single striatal neurons project simultaneously to GPe, the internal globus pallidus, and SNr in primates and rodents[53–55]. Our anatomical work has also shown that some CDt neurons were double-labeled with retrograde tracers injected into cvGPe and cdlSNr[24]. These imply that the direct and indirect pathways may not completely be segregated at the single striatal neuron level. However, the present study has demonstrated that cdlSNr and cvGPe neurons receive the stable value signals from CDt in an opposite manner (see Figs. 3e and 4e, Supplementary Fig. 4a–c), and that behavioral alternations are induced by the optical stimulation of the CDt-cdlSNr pathway, but not the CDt-cvGPe pathway (see Fig. 6d–i, Supplementary Fig. 8). These lines of experimental evidence are directed toward an indication that the direct and indirect pathways lie in an at least functionally independent fashion. In favor of this notion, it has recently been shown that optogenetic manipulation targeting D1 vs. D2 dopamine receptor-expressing striatal neurons produces distinct behavioral effects[56–59], although both receptors have been revealed to co-exist in a certain population of MSNs[60–62].

The present study defines that the combination of task-related neuronal activity recording and pathway-selective optogenetic manipulation is a powerful technique for analyzing the operative mechanisms and functional roles of neuronal networks in the primate brain. For example, this approach can be applied to other pathways in the primate basal ganglia. In addition to the CDt-cdlSNr-SC network examined here, a parallel saccade-related network originates from the head of the caudate nucleus (CDh)

3,18,22,39,63. This network has been reported to encode predictive or flexible values of visual objects[18,22] and may contribute to saccadic gaze shifts through a similar disinhibitory mechanism. Further investigations are warranted to elucidate how the network arising from the CDh conveys value signals to SC through SNr for the modulation of saccade signals.

## Methods

**Subjects**. Two adult male rhesus monkeys (*Macaca mulatta*, monkeys SH and ZB) were used for the present study. All animal care and experimental procedures were approved by the National Eye Institute Animal Care and Use Committee and complied with the Public Health Service Policy on the Humane Care and Use of Laboratory Animals. We implanted a plastic head holder, plastic recording chambers, and scleral search coils under general anesthesia.

**Stimuli**. We created visual stimuli using fractal geometry[26]. One fractal object was composed of four point-symmetrical polygons that were overlaid around a common center such that smaller polygons were positioned more toward the front. The parameters that determined each polygon (size, edge, color, etc.) were chosen randomly. Fractal objects' sizes were on average ~8°×8° but ranged from 5° to 10°.

**Experimental control**. Behavioral procedures were controlled by custom-made C++-based software Blip[64]. Data acquisition and output control were performed using a Multifunction I/O Device (NI-PCIe 6353; National Instruments). The subject sat in a primate chair with its head fixed facing a fronto-parallel screen 30 cm from subject's eyes in a sound-attenuated and electrically shielded room. Visual stimuli generated by an active-matrix LCD projector (PJ658; ViewSonic) were rear-projected on the screen. Diluted apple juice (33%) was used as reward. The juice rewards were delivered through a computer-controlled electromagnetic solenoid valve (Parker-Hannifin). Eye position was sampled at 1 kHz using a search coil-based eye tracker system (DNI Instruments).

Object-reward association task (Figs. 1b and 5f): each subject learned the association of fractal objects with larger or smaller reward in the object-reward association task[34]. Each session was performed with a set of eight objects (four good and four bad). In training sessions, after a period of fixation on a central dot, one object appeared on the screen in a pseudo-random manner at one of the eight peripheral locations (eccentricity 15°). After an overlap period of 400 ms, the fixation dot disappeared and the subject was required to make a saccade to the object. After 600 ms of gazing at the object, either a large or small reward was delivered with a correct tone. Each session consisted of 64 trials. In recording sessions when we investigated saccadic latency (Fig. 5g) and neuronal activity in cdlSNr and SC (Fig. 5h), the procedure was the same except that objects were presented at a single location chosen to match the neuron's receptive field (Fig. 5f).

Passive viewing task (Fig. 1c): we used the passive viewing task to examine whether neurons encoded the stable values of the objects[34]. Although the subject maintained fixation on a central dot, some of the learned objects (one to six per trial) were sequentially presented at a same neuron's preferred location in a pseudo-random manner. The duration of each object presentation was 400 ms. A fixed reward was delivered with a correct tone 300 ms after the last object presentation irrespective of good or bad object values.

**Object free-viewing task**. We used the object free-viewing task to examine whether the subject learned the stable values of objects[20]. An array of four objects (good and bad, pseudo-randomly selected from a set of eight) was presented on the screen for 3 s, and the subject was free to look at or ignore the objects during this time. After array offset a white fixation dot appeared, and a reward was delivered if the subject maintained fixation on the white dot for 1 s. The reward volume delivered on each trial was unaffected by the value of preceding objects.

Movie free-viewing task (Fig. 6a): we used the movie free-viewing task to examine the effect of optical stimulation on neurons and otherwise uncontrolled behavior. The subject freely watched either natural history movies from online database (ARKive; www.arkive.org) during the optical stimulation experiment. A small reward was occasionally delivered at random for the sake keeping the subject awake. Optical stimulation (with laser light) or dummy stimulation (without laser light) was delivered pseudo-randomly irrespective of eye position or movie content. The stimulation duration was 20 ms, and inter-trial intervals were 3–5 s.

**Single-unit recordings**. We recorded neuronal activity using an epoxy-coated tungsten microelectrode (0.2–2.0 MΩ, 200 μm thick; FHC). For monkey SH, we placed two rectangular chambers, one right lateral chamber tilted laterally by 25° targeting the right CDt, cdlSNr, and cvGPe and the other posterior chamber tilted posteriorly by 40° targeting both sides of the SC based on a stereotaxic atlas. For monkey ZB, we placed two rectangular chambers, one left lateral chamber tilted by 15° targeting the left CDt, cdlSNr, and cvGPe and the other posterior chamber tilted posteriorly by 40° targeting both sides of the SC. MR images (4.7 T, Bruker) were obtained aligned with the direction of the recording chamber, which was visualized with gadolinium that filled the grid holes and inside the chamber. Single-

unit recording was performed using epoxy-coated (200 μm thick; FHC) or glass-coated (350 μm thick; Alpha-Omega) tungsten electrodes. The recording sites were targeted using a grid system with 1 mm spacing. Based on the MR images and preceding recording data, we chose a grid hole to hold the stainless-steel guide tube, through which the electrode was inserted and advanced by an oil-driven micro-manipulator (Narishige). The neuronal electrical signal was amplified, bandpass filtered (200 Hz to 10 kHz; BAK), and collected at 40 kHz. Spikes from single neurons were isolated online using a custom voltage-time window discrimination software application in Blip.

We limited the number of neurons recorded in CDt (total 21 neurons) using the optrode, because otherwise CDt-related fiber connections could be damaged profoundly by repetitive recordings from CDt. Though we sampled all types of CDt neurons to test the optical stimulation effect, the sampling bias of neuron types reported is ascribable to higher spontaneous firing rates of FSIs and TANs than those of MSNs.

For simultaneous recordings in both cdlSNr (by optrode) and SC (by recording electrode), we determined the SC recording site based on the receptive field map of SC after checking the receptive field of cdlSNr neurons.

**Viral transfection**. We used a custom-made injectrode consisting of a tungsten microelectrode (200 μm thick; FHC Inc.) and a silica tubing (outer/inner diameter: 150/75 μm; Polymicro technologies) for single-unit recording and vector injection. The distance between the tips of the electrode and the beveled silica tubing was ~ 500 μm. To locate the injection sites, we recorded neuronal activity using the injectrode before viral injections, because the injections would be unlikely to target CDt successfully without guidance of electrophysiological recording, as CDt is an elongated but thin structure (about 1 mm wide). We injected vector into four sites in CDt for each subject (Fig. 1e). As viral vector, we used AAV2-CMV-ChR2 (H134R)-EYFP ($8.5 \times 10^{12}$ genome copy ml$^{-1}$)[27]. A total volume of 4.0 μL (monkey SH) or 3.0 μL (monkey ZB) of the vector was injected at each site. We first injected 0.1 μL of vector at a speed of 0.4 μL min$^{-1}$, followed by 3.9 μL (monkey SH) or 2.9 μL (monkey ZB) at a speed of 0.08 μL min$^{-1}$ using a 10 μL Hamilton syringe with a 30-gauge stainless-steel needle held in a motorized infusion pump (Harvard Apparatus). Vector was injected in four sites along CDt (12, 14, 16, and 18 mm posterior to AC) in the right hemisphere of SH, and four sites along CDt (10, 12, 14, and 16 mm posterior to AC) in the left hemisphere of ZB.

**Optical stimulation**. We used a custom-made optrode consisting of a tungsten microelectrode (125 μm thick; FHC Inc.) and a fiber-optic cable (200 μm core diameter; Doric Lenses) for single-unit recording and optical stimulation. The distance between the tips of the electrode and the optical fiber was 250–500 μm. We used a 473-nm DPSS blue laser light for the light source (IKE-S-473-500-100-T; IkeCool Cooperation). We left the laser continuously on during the experiment, and switched the light stimulation on and off by a mechanical shutter placed in the light path (CO12-M4-FC-1-5H-L; Luminos). Before each experiment we measured the light intensity from the tip of the optrode using an optical power meter (1916-C; Newport Cooperation) coupled with an 818-SL/DB photo detector[65]. We used 32–955 mW mm$^{-2}$ light for the stimulation.

**Histology**. After completing all experiments, both subjects were deeply anesthetized with an overdose of sodium pentobarbital (390 mg ml$^{-1}$) and perfused transcardially with saline followed by 4% paraformaldehyde. The head was fixed to the stereotaxic frame and the brain was cut into a block in the coronal plane including the midbrain region. The block was post-fixed overnight at 4 C°, and then cryoprotected for one week in increasing gradients of glycerol solution (10–20% glycerol in phosphate-buffered saline; PBS) before being frozen. Frozen blocks were cut every 50 μm using a microtome. Slices taken at 250 μm-intervals were used for enhancing the EYFP signals by immunofluorescence, and the adjacent slices were used for Nissl staining.

**Immunofluorescence**. For enhancing the EYFP signals, we used rabbit anti-GFP antibody (G10362; ThermoFisher) and goat anti-rabbit IgG antibody conjugated with alexa fluor 488 (A-11034; ThermoFisher). The sections were preincubated for 30 min in 0.3% hydrogen peroxide in 0.1 M PBS (pH 7.4) to block endogenous peroxidase, followed by three rinses through 0.1 M PBS, followed by 1 h in blocking solution containing 5% normal goat serum in 0.1 M PBS. The sections were incubated for 18 hours at room temperature in blocking solution containing 2.5% normal goat serum and 0.1% TX-100 with rabbit anti-GFP antibody (1:2000). After three rinses with PBS, the sections were incubated for 2 h at room temperature with goat anti-rabbit IgG antibody conjugated with alexa fluor 488 (1:200). We captured the fluorescent images of the labeled neurons using a fluorescence microscope (BZ-X700; Keyence). We adjusted the contrast and brightness of each color channel using Photoshop (Adobe) to enhance the ability to differentiate fluorescently labeled neurons.

**Data analyses**. All neurophysiological and behavioral data analyses were performed using MATLAB R2018b (Mathworks). To assess whether the neurons represented stable values, we compared the neuronal responses to good objects with bad objects in a statistical test window after the object onset in the passive

viewing task. The statistical test windows were set as 100–400 ms after the object onset for CDt (Fig. 1d), cdlSNr (Fig. 3c, e, h), and cvGPe neurons (Fig. 4c, e, h) based on a previous work[16]. For investigating whether the neurons showed the value representation after object onset and before saccade onset, we compared the neuronal responses to good objects with bad objects in two test windows in the object-reward association task. The test window for the post-period of object onset was set as 100–300 ms after the object onset to exclude the pre-saccadic activity, and the test window for the post-period of fixation offset was set as 400–500 ms after the fixation offset to include the saccadic activity for both good and bad objects for cdlSNr (Fig. 5h, middle) and SC neurons (Fig. 5h, bottom). The stable value index was defined as the area under the receiver operating characteristic (ROC) curve based on the response magnitudes to good objects vs. bad objects (Fig. 3f, i, Fig. 4f, i)[16,19].

For investigating neuronal responses to the optical stimulation, we compared neuronal activities in a post-stimulation test window with a baseline window (0–100 ms before the stimulation onset) for each neuron. We set the test window to 5–15 ms for cdlSNr neurons (Fig. 3b, d–g), 5–10 ms for cvGPe neurons (Fig. 4b, d–g), 0–100 ms for SC neurons based on the different response patterns of each region (Fig. 5c–e). Responses to optical stimulation were deemed significant if a Wilcoxon signed-rank test yielded a P value of <0.05.

For investigating the latency of neuronal responses in cdlSNr and cvGPe to optical stimulation (Supplementary Fig. 2), the latency was calculated by a bootstrap analysis. We set the time window of 200 ms period before stimulation (pre-period) and sliding test window (5 ms duration; slid by 1 ms) starting after the stimulation (post-period). We compared the average firing rate of the pre-period with the post-period. Fifty trials were randomly resampled for a new bootstrap data set. The comparison with the random sampling data was repeated 1000 times. If the average firing rate of the post-period was lower or higher than the pre-period in >975 repetitions, the response within that test window was deemed significant. The left-most edge of four consecutive significant responses was defined as the response latency. We excluded the data when the bootstrap analysis failed to detect a significant response.

For investigating the saccade latency in the object-reward association task (Fig. 5g), we compared the saccade latency toward good objects with bad objects.

For investigating the behavioral effect by the optical stimulation, we compared the number of all saccades (Fig. 6f, g and Supplementary Fig. 8c, d) and the first saccades (Supplementary Fig. 7a, b) in 0–200 ms after the stimulation onset in optical stimulation trials with no-stimulation trials in each session.

**Statistics**. Statistical analyses were completed using MATLAB R2018b (Math-Works). For analyzing the cdlSNr and SC neuron activities in Fig. 5h, we applied a parametric test (two-sided, paired t test). We confirmed that the parametric test was applicable for these neurophysiological data by two-sample Kolmogorov–Smirnov test and two-sample F test. Other neurophysiological and behavioral data were analyzed using a non-parametric test (Mann–Whitney U test or Wilcoxon signed-rank test). Neurophysiology results were reported as mean (±SEM).

**Reporting summary**. Further information on research design is available in the Nature Research Reporting Summary linked to this article.

## Data availability
The data sets generated and/or analyzed during the current study are available from the corresponding author (H.A.) on reasonable request. The source data underlying Figs. 3–6 and Supplementary Figs. 4 and 8 are provided as a Source Data file.

## Code availability
The MATLAB (MathWorks) code used for data analysis is available from the corresponding author.

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

## Acknowledgements
This work was supported by the National Eye Institute Intramural Research Program at the National Institutes of Health, JST PRESTO Grant JPMJPR1683 (K.I.) and JST CREST Grant JPMJCR1853 (M.T.). We thank M.K. Smith for histological expertize and service; A. Ghazizadeh, J. Cavanaugh, M. Nakano, and M. Fujiwara for technical assistance and advice; D. Parker, I. Bunea, D. O'Brien and G. Tansey for help with animal care and surgery; A.M. Nichols, T.W. Ruffner, and A.V. Hays for providing equipment; D. Yu and F.Q. Ye for MRI services; D. McMahon for comments that greatly improved the manuscript.

## Author contributions
H.A., H.F.K., and O.H. conceived and designed experiments. K.I and M.T. produced the virus vectors. H.A., H.F.K., and K.I. performed experiments. H.A. analyzed data and made figures. H.A., M.T., and O.H. wrote the manuscript. All authors discussed the results and edited the manuscript.

## Competing interests
The authors declare that they have no competing interests.
