## [Peer Review File · Nature Communications]

Reviewers' Comments:

Reviewer #1:

Remarks to the Author:

The manuscript of Amita et al. reports the results of a pathway-selective manipulation based on viral-mediated gene transfer in the brain of macaque monkeys. The authors focused on output pathways from the so-called « tail of the striatum » (including the caudate tail, CDt) to specific parts of the substantia nigra pars reticulata (SNr) and external segment of the globus pallidus (GPe), and from there to superior colliculus. They examined the physiological impact of each pathway activation and its causal effect on gaze control. Based on previous work in the Hikosaka lab, it is known that CDt neurons emit stable value signals that have been acquired through long-term experience with visual stimuli associated with "good" or "bad" outcomes, thus leading to automatic gaze orientation toward "good" stimuli. Using precisely-targeted gene manipulation of neuronal pathways previously employed in the Inoue lab, the authors show that optogenetic stimulation of CDt axon terminals to the SNr and GPe has consequences on oculomotor behavior. Given the relative paucity of reports using the optogenetic approach in behaving monkeys, this study may help to extend our understanding of the circuit-level interactions that mediate behavior in the basal ganglia. In this respect, the originality of the current study is undisputable. The data as a whole are interesting and clearly presented. All figures are well conceived and useful (with one exception, as stated in Minor comments).

Major comment

(1) The injected viral vector was conveyed to the CDt axon terminals within the SNr and GPe where channelrhodopsin-2 was expressed. This procedure allowed the authors to assess the effects of optically stimulating the two primary output populations of the striatum, the "direct CDt pathway" (in the SNr) and "indirect CDt pathway" (in the GPe), while monkeys were performing visual tasks. Such an approach is important for the field of basal ganglia physiology to study separately the two pathways originating from distinct populations of projection neurons in the striatum. Although the current findings provide evidence for a competitive functioning of direct and indirect CDt pathways during oculomotor behavior (SNr and GPe neurons were inhibited by good and bad stimuli, respectively), neuroanatomical data in the literature lead to put into perspective the notion of a dual striatofugal system. Indeed, it has been documented that some striatal output neurons project to multiple downstream structures of the basal ganglia. As emphasized by Lévesque and Parent, a majority of striatofugal neurons in primates arborize in the three major target structures of the striatum (i.e., GPe, GPi, and SNr) through collateralized axons. The authors of the present study themselves have shown that a proportion of neurons from the striatum tail apparently project to both SNr and GPe (Amita et al. 2019, EJM paper). This suggests that striatofugal pathways might not be as segregated as classically considered. It would be interesting if the authors could comment on this point.

Minor comments

(1) In the Methods, the authors say « We first injected 0.1 μ L of vector at a speed of 0.4 μ L/min, followed by 3.9 μ L (monkey SH) or 2.9 μ L (monkey ZB) at a speed of 0.08 μ L/min ». Does that mean that a 3-4 μ L injection of viral vector was made at each site ? Please, make it clearer in the text.

(2) On p.6, the authors state that a postmortem analysis was used to verify transgene expression in CDt neurons in monkey SH. But, as mentioned later (p.20), both monkeys were euthanized. Has the histological analysis been carried out on only one monkey or both of them ?

(3) Could you give us further information on how the time window analysis is conducted (p.21), in particular, for choosing the test window durations where the spiking activity is measured ?

(4) Fig. 6B is a bit muddled. I found it uneasy to see a link between saccade characteristics (i.e., ipsi and contralateral, including their angle and length) and spiking activity on this example neuron.

Reviewer #2:

Remarks to the Author:

The authors applied pathway-selective optogenetic manipulation to elucidate how the good and bad value of information modulates the activity of the basal ganglia-superior colliculus system in saccade execution. The authors demonstrated that CDt facilitates saccades toward good objects by serial inhibitory pathways through SNr.

This is an interesting, well-written paper, which could be suitable for publication in Nature Communication after a moderate revision. The experiments are well documented and mostly clear. However, some technical and theoretical issues have to be cleared in the revision of the manuscript.

Major points:

1. There is no clear aim of the study at the end of the introduction. The last paragraph of the introduction is not appropriate to the introduction. Instead of this part, which describes the main findings a clear aim of the study and its justification would be helpful for the readers.
2. The first point of the results: 'Injecting vector into CDt for expressing opsin' is more connected to the methods part than to the results. Replace it to the methods.
3. After vector injection in the CDt the authors checked 5 MSN 10 TAN and 5 FSI neurons in the CD. This distribution is far from the distribution of these different neuron types in the CDt (MSN: more than 75%). Why was chosen this CDt neuron sample of this test?
4. The proportion of the infected neurons was also not similar among these three groups. Why? It is clear that the labeling is not selective but based on the demonstrate results it is hard to say that different neuron types were labelled equally.
5. Sometimes t-test, sometimes Mann-Whitney test sometimes Wilcoxon test. Parametric, non-parametric, paired, non-paired test are applied without the justification of the reliability of the applied tests. Was normality check test, variance homogeneity test performed before the above-mentioned test? If yes, please add this information to the results. If not please do the pre-tests and repeat the statistical analysis.
6. Line 147-148: The authors stated that: 'we turned to the cvGPe? Why? Why not the subthalamic nucleus? This is a part of the indirect pathway of the basal ganglia, too.
7. Lines 178-185: This speculation is not an essential part of the result. Remove if or put it to the discussion.
8. It is stated in the line 195: In this configuration, we first investigated the responses of a cdISNr neuron and a SC neuron to the optical stimulation in cdISNr. Were the SNr neurons stimulated? Or the terminals from the CDt as described earlier in the manuscript.
9. It can be read in the discussion: Accordingly, the information concerning a good object presented at a given position might be conveyed topographically along the CDt-cdISNr-SC network in which individual neurons have similar receptive fields. Did the authors mapped the receptive fields in the CD, SNr and SC? It was described earlier (Pouderoux and Freton 1979 Neurosci Lett; Nagy et al, 2003, 2006, Eur J Neurosci) that in the SNr and the CN of the feline brain large receptive fields can be found, which covers practically the whole visual field of the investigatd eye. What was the case in the monkey?
On the other hand, the receptive fields in the SCi are much smaller. How could the authors make the overlap between the receptive fields?

Minor points:

1. During the categorization of the CDt neurons did the authors checked the shape of the

- autocorrelograms? This is an important aspect of the categorization in the CD, too
2. Line 193: which intermediate SC layer?
 3. 247-248: after optical stimulation... What was stimulated here?

Reviewer #3:

Remarks to the Author:

Amita and colleagues used pathway specific optogenetics to determine the contribution the direct and indirect pathways from the caudate tail played in generating saccadic eye movements. In line with previous results using orthodromic stimulation, optogenetic stimulation of the CDt causes an inhibition of neurons in the SNr and GPe and preferentially causes inhibition is stable value coding neurons in these nuclei. Furthermore, optogenetic stimulation of the CDt terminals in the SNr caused an increase in firing in the superior colliculus and a facilitation of saccadic eye movements. Interestingly there was no effect of CDt terminal stimulation in the GPe on either SC neuronal activity or in the direction of saccadic eye movements.

Overall the results are well described and the data adds nicely to the previous work from this lab about stable value coding in the CDt pathway.

We just have a few minor points that we would like more clarification on. These minor points do not largely affect the quality of the study but should be clarified.

- The authors only recorded 7 SC units while stimulating CDt terminals in the GPe. None of these neurons were responsive to optogenetic stimulation of CDt fibers in the GPe. The authors did not report whether optogenetic stimulation of the CDt fibers in the GPe lead to a change in firing in the SNr. If this optogenetic stimulation did not result in a change in firing in the SNr then there would be no expected change in the activity of the SC units. Have the authors recorded in the SNr while stimulating the CDt fibers in the GPe? If they have or have not can they comment because this will affect the interpretation of the negative results obtained with stimulating the CDt fibers in the GPe. Essentially are these results a technical failure in that the SNr firing is not affected or is the SNr firing affected and the model for how this will impact the SC wrong?

- In both the GPe and the SNr there were both positive and negative value neurons that were sensitive to optogenetic stimulation. Can these be separated with examples of each shown in the supplemental information. This could be important as the population response presented in 3e does not appear to be significantly inhibited by either good or bad objects. Presumably this is due to a small number of positive coding neurons influencing the population response. If it is not due to the positive coding neuron and the negative coding population does not on average look like the example in 3c this needs to be discussed as it would not, then reflect the model they have previously and currently propose in 7. It would help in this regard if there was an analysis to show what proportion of neurons were significantly inhibited (below baseline) during the statistical test window.

Reviewers' comments:

Reviewer #1 (Remarks to the Author):

The manuscript of Amita et al. reports the results of a pathway-selective manipulation based on viral-mediated gene transfer in the brain of macaque monkeys. The authors focused on output pathways from the so-called « tail of the striatum » (including the caudate tail, CDt) to specific parts of the substantia nigra pars reticulata (SNr) and external segment of the globus pallidus (GPe), and from there to superior colliculus. They examined the physiological impact of each pathway activation and its causal effect on gaze control. Based on previous work in the Hikosaka lab, it is known that CDt neurons emit stable value signals that have been acquired through long-term experience with visual stimuli associated with "good" or "bad" outcomes, thus leading to automatic gaze orientation toward "good" stimuli. Using precisely-targeted gene manipulation of neuronal pathways previously employed in the Inoue lab, the authors show that optogenetic stimulation of CDt axon terminals to the SNr and GPe has consequences on oculomotor behavior. Given the relative paucity of reports using the optogenetic approach in behaving monkeys, this study may help to extend our understanding of the circuit-level interactions that mediate behavior in the basal ganglia. In this respect, the originality of the current study is undisputable. The data as a whole are interesting and clearly presented. All figures are well conceived and useful (with one exception, as stated in Minor comments).

Major comment

(1) The injected viral vector was conveyed to the CDt axon terminals within the SNr and GPe where channelrhodopsin-2 was expressed. This procedure allowed the authors to assess the effects of optically stimulating the two primary output populations of the striatum, the "direct CDt pathway" (in the SNr) and "indirect CDt pathway" (in the GPe), while monkeys were performing visual tasks. Such an approach is important for the field of basal ganglia physiology to study separately the two pathways originating from distinct populations of projection neurons in the striatum. Although the current findings provide evidence for a competitive functioning of direct and indirect CDt pathways during oculomotor behavior (SNr and GPe neurons were inhibited by good and bad stimuli, respectively), neuroanatomical data in the literature lead to put into perspective the notion of a dual striatofugal system. Indeed, it has been documented that some striatal output neurons project to multiple downstream structures of the basal ganglia.

As emphasized by Lévesque and Parent, a majority of striatofugal neurons in primates arborize in the three major target structures of the striatum (i.e., GPe, GPi, and SNr) through collateralized axons. The authors of the present study themselves have shown that a proportion of neurons from the striatum tail apparently project to both SNr and GPe (Amita et al. 2019, EJM paper). This suggests that striatofugal pathways might not be as segregated as classically considered. It would be interesting if the authors could comment on this point.

Reply: As the reviewer commented, previous anatomical studies suggest that single striatal neurons project to both SNr and GPe. We agree that it is better to refer to this topic in Discussion. We have added one paragraph in Discussion as below (lines 338-350, p. 17-18).

“Anatomical studies have reported that single striatal neurons project simultaneously to GPe, the internal globus pallidus, and SNr in primates and rodents. Our anatomical work has also shown that some Cdt neurons were double-labeled with retrograde tracers injected into cvGPe and cdLSNr. These imply that the direct and indirect pathways may not completely be segregated at the single striatal neuron level. However, the present study has demonstrated that cdLSNr and cvGPe neurons receive the stable value signals from Cdt in an opposite manner (see Fig. 3e,4e, Supplementary Fig. 4a-c), and that behavioral alternations are induced by the optical stimulation of the Cdt-cdLSNr pathway, but not the Cdt-cvGPe pathway (see Fig. 6d-i, Supplementary Fig. 8). These lines of experimental evidence are directed toward an indication that the direct and indirect pathways lie in an at least functionally independent fashion. In favor of this notion, it has recently been shown that optogenetic manipulation targeting D1 vs. D2 dopamine receptor-expressing striatal neurons produces distinct behavioral effects, although both receptors have been revealed to co-exist in a certain population of MSNs.”

Minor comments

(1) In the Methods, the authors say « We first injected 0.1 μ L of vector at a speed of 0.4 μ L/min, followed by 3.9 μ L (monkey SH) or 2.9 μ L (monkey ZB) at a speed of 0.08 μ L/min ». Does that mean that a 3-4 μ L injection of viral vector was made at each site? Please, make it clearer in the text.

Reply: Yes, this part means that a total volume of 3-4 μ L of the vector was injected at each site. To make it clearer, we have added one sentence in ‘Viral transfection’ of Methods as below (lines 442-443, p. 22).

“A total volume of 4.0 μ L (monkey SH) or 3.0 μ L (monkey ZB) of the vector was

injected at each site.”

(2) On p.6, the authors state that a postmortem analysis was used to verify transgene expression in CDt neurons in monkey SH. But, as mentioned later (p.20), both monkeys were euthanized. Has the histological analysis been carried out on only one monkey or both of them?

Reply: We performed the histological analysis in both monkeys (monkeys SH and ZB), and found ChR2-EYFP expression in CDt cell bodies, and CDt terminals in cdlSNr and cvGPe in both monkeys. To indicate that the results obtained in monkey SH are representative, we have revised the corresponding part as below (lines 102-103, p. 6).

“Histological results also confirmed the expression of ChR2-EYFP in both monkeys. Representative sections obtained in monkey SH are depicted in Figure 2d-f.”

(3) Could you give us further information on how the time window analysis is conducted (p.21), in particular, for choosing the test window durations where the spiking activity is measured?

Reply: We have added the further information about how we chose the test window durations as below (lines 484-490, p. 24).

“... based on a previous work. For investigating whether the neurons showed the value representation after object onset and before saccade onset, we compared the neuronal responses to good objects with bad objects in two test windows in the object-reward association task. The test window for the post-period of object onset was set as 100–300 ms after the object onset to exclude the pre-saccadic activity, and the test window for the post-period of fixation offset was set as 400–500 ms after the fixation offset to include the saccadic activity for both good and bad objects for cdlSNr (Fig. 5h, middle) and SC neurons (Fig. 5h, bottom).”

(4) Fig. 6B is a bit muddled. I found it uneasy to see a link between saccade characteristics (i.e., ipsi and contralateral, including their angle and length) and spiking activity on this example neuron.

Reply: Except for Figure 6b, we show the data using common methods (Fig. 6c-i). These graphs are based on various sets of data, including saccade timing, direction, and amplitude, temporal change of saccade direction, etc. In most cases, we separated these data into two groups (e.g., without vs. with optical stimulation, before vs. after the stimulation, contralateral vs. ipsilateral direction). It was then difficult to check the relationships between various sets of data, for example, each saccade (timing, direction,

amplitude) vs. each action potential of the SC neuron, especially in each trial. Such many sets of data correlations can be provided in Figure 6b, some of which might be fundamentally but unexpectedly important. This is what we intended to do.

Reviewer #2 (Remarks to the Author):

The authors applied pathway-selective optogenetic manipulation to elucidate how the good and bad value of information modulates the activity of the basal ganglia-superior colliculus system in saccade execution. The authors demonstrated that CDt facilitates saccades toward good objects by serial inhibitory pathways through SNr.

This is an interesting, well-written paper, which could be suitable for publication in Nature Communication after a moderate revision. The experiments are well documented and mostly clear. However, some technical and theoretical issues have to be cleared in the revision of the manuscript.

Major points:

1. There is no clear aim of the study at the end of the introduction. The last paragraph of the introduction is not appropriate to the introduction. Instead of this part, which describes the main findings a clear aim of the study and its justification would be helpful for the readers.

Reply: We totally agree that we should describe the aim of the study, main findings, and their justification in the last paragraph of Introduction. We have revised the corresponding part in Introduction as below (lines 60-68, p. 4).

“To examine how the direct and indirect pathways from CDt encoding the stable value modulate saccadic eye movements, we optogenetically activated either one of the pathways. This was achieved by optical stimulation in either cdLSNr or cvGPe of the axon terminals of CDt neurons. We found that the selective activation of the direct CDt-cdLSNr pathway conveying stable value signals facilitated contralateral saccades through the disinhibition of SC, whereas the activation of the indirect CDt-cvGPe pathway showed no significant effect on saccades. Therefore, we conclude that the multisynaptic CDt-cdLSNr-SC pathway encoding the stable value plays a crucial role in generation of saccades toward good objects within the contralateral hemifield.”

2. The first point of the results: ‘Injecting vector into CDt for expressing opsin’ is more connected to the methods part than to the results. Replace it to the methods.

Reply: We also agree that ‘Injecting vector into CDt for expressing opsin’ should be connected to Methods. Thus, we have changed the title to ‘Opsin expression in CDt neurons’ and moved the following part of this section to ‘Viral transfection’ of Methods (lines 437-440, p. 22).

“To locate the injection sites, we recorded neuronal activity using the injectrode before

viral injections, because the injections would be unlikely to target CDt successfully without guidance of electrophysiological recording, as CDt is an elongated but thin structure (about 1 mm wide).”

3. After vector injection in the CDt the authors checked 5 MSN 10 TAN and 5 FSI neurons in the CD. This distribution is far from the distribution of these different neuron types in the CDt (MSN: more than 75%). Why was chosen this CDt neuron sample of this test?

Reply: There are several possibilities of the reason why our sampling was far from the neuronal proportions based on previous anatomical studies. First, it is very easy to detect FSIs and TANs as compared to MSNs, because FSIs and TANs fire tonically whereas the spontaneous firing rate of MSNs is very low. Second, a population of neurons is affected by the recording condition. According to our experience, we sampled more interneurons using lower-impedance electrodes, while we sampled more MSNs using higher-impedance electrodes. In this study, we repetitively used the same optrodes for recording and stimulation, and their impedance became lower after several penetrations. This is another reason why we sampled more interneurons. Third, our main goal was to examine the neuronal circuits originating from CDt. For this purpose, we limited the number of neurons recorded in CDt, because otherwise CDt-related fiber connections could be damaged profoundly by repetitive recordings from CDt using the optrode. We have added some descriptions in Methods as below (lines 426-430, p. 21-22).

“We limited the number of neurons recorded in CDt (total 21 neurons) using the optrode, because otherwise CDt-related fiber connections could be damaged profoundly by repetitive recordings from CDt. Though we sampled all types of CDt neurons to test the optical stimulation effect, the sampling bias of neuron types reported is ascribable to higher spontaneous firing rates of FSIs and TANs than those of MSNs.”

4. The proportion of the infected neurons was also not similar among these three groups. Why? It is clear that the labeling is not selective but based on the demonstrate results it is hard to say that different neuron types were labelled equally.

Reply: This is a very important comment. As the reviewer pointed out, the proportion of MSNs expressing ChR2 was less than those of TANs and FSIs. This may be due to the difference in transduction efficiency among the cell types. In addition, the lack of MSNs may be explained by an indirect effect through other excited interneurons in CDt. However, it should be noted that the number of CDt neurons was small, as our above

reply to the Reviewer's Comment 3. To address this issue, it is necessary to record from a number of neurons in CDt. We have added some descriptions to mention it as below (lines 96-100, p. 6).

“The different proportions of neurons responding to the optical stimulation may be due to the difference in the transduction efficiency among cell types. In addition, the lack of excitation in some MSNs (Supplementary Fig. 1c, Excitation (-)) may be caused by a possible indirect effect through excited FSIs and other interneurons in CDt (Fig. 2a).”

5. Sometimes *t*-test, sometimes Mann-Whitney test sometimes Wilcoxon test. Parametric, non-parametric, paired, non-paired test are applied without the justification of the reliability of the applied tests. Was normality check test, variance homogeneity test performed before the above-mentioned test? If yes, please add this information to the results. If not please do the pre-tests and repeat the statistical analysis.

Reply: Thank you for the suggestion. We did not perform a normality check test or a variance homogeneity test before the paired *t*-test. As this reviewer suggested, we confirmed that the paired *t*-test was applicable by the two-sample Kolmogorov-Smirnov test and the two-sample *F*-test for comparing responses to good objects with those to bad objects in paired SNr and SC neurons (Fig. 5h). And, we have added the information on the *F*-test in Results (p. 11-12) as below.

Lines 227-228, p. 11

“Fig. 5h, middle-left, $F(4, 4) = 0.57, P = 0.60, F\text{-test}; P^{**} = 0.0045, \text{paired } t\text{-test}$ ”

Lines 229-230, p. 12

“Fig. 5h, bottom-left, $F(4, 4) = 1.46, P = 0.72, F\text{-test}; P^{**} = 0.0083, \text{paired } t\text{-test}$ ”

Line 236, p. 12

“Fig. 5h, middle-right, $F(4, 4) = 0.72, P = 0.75, F\text{-test}; P^* = 0.021, \text{paired } t\text{-test}$ ”

Line 238, p. 12

“Fig. 5h, bottom-right, $F(4, 4) = 0.97, P = 0.98, F\text{-test}; P^* = 0.017, \text{paired } t\text{-test}$ ”

We applied a non-parametric test (Mann-Whitney *U* test or Wilcoxon signed-rank test) for other neurophysiological and behavioral data. We re-analyzed the neuron data using the non-parametric test (Wilcoxon signed-rank test) for investigating neuronal responses to the optical stimulation. Based on this re-analysis, we have revised Results and Figures (Figs. 2-5). Conclusion was not affected by this re-analysis. In addition, we have revised the corresponding part in “Statistics” of Methods as below (lines 515-520, p. 25-26).

“For analyzing the cdLSNr and SC neuron activities in Figure 5h, we applied a parametric test (two-sided, paired *t*-test). We confirmed that the parametric test was applicable for these neurophysiological data by two-sample Kolmogorov-Smirnov test and two-sample *F*-test. Other neurophysiological and behavioral data were analyzed using a non-parametric test (Mann–Whitney *U* test or Wilcoxon signed-rank test).”

6. Line 147-148: The authors stated that: ‘we turned to the cvGPe? Why? Why not the subthalamic nucleus? This is a part of the indirect pathway of the basal ganglia, too.

Reply: It was previously shown that the indirect pathway consists of a sequential connection through the subthalamic nucleus (STN): CD-GPe-STN-SNr/GPi. But, more recent studies have shown that GPe has direct connections to SNr/GPi (Smith and Bolam, Neuroscience, 1998). Indeed, our previous studies based on anatomical and electrophysiological data showed that the indirect pathway of CDt is composed of two sequential inhibitory connections (i.e., CDt-cvGPe-cdLSNr) (Kim et al., Neuron, 2017; Amita et al., Eur. J. Neurosci., 2019), although it is still possible that STN is partially involved. In any case, cvGPe, rather than STN, should be the important region for the optogenetic experiment because the optical stimulation would manipulate only the direct target of CDt neurons (which can be cvGPe, not STN). Thus, we have revised the corresponding part to explain why we investigated cvGPe (lines 158-159, p. 8).

“To explore this issue further, we turned to cvGPe, which is another projection site of CDt (Fig. 2f) and has inhibitory neuron responses to bad objects.”

Nonetheless, the reviewer’s comment turned out to be very important to understand the unclear data about the optical stimulation in cvGPe (unlike cdLSNr). One possibility is that the optical stimulation inhibits cvGPe neurons, but also affects other neurons including STN neurons which might disrupt the direct effect on the cvGPe-cdLSNr projection. Thus, we have added this possibility in Discussion (lines 332-337, p. 17).

“This might be ascribed to the complex pattern of heterogeneous GPe neuron connectivity. In addition to the indirect pathways described above (CDt-cvGPe-(STN)-cdLSNr pathways), cvGPe neurons may have reciprocal connections with STN neurons or striatal neurons, or have mutual inhibitory connections within GPe. Thus, the optical stimulation might reduce the presumed effect on the indirect pathways.”

7. Lines 178-185: This speculation is not an essential part of the result. Remove if or put it to the discussion.

Reply: As the reviewer suggested, we have removed the corresponding part from the manuscript.

8. It is stated in the line 195: In this configuration, we first investigated the responses of a cdLSNr neuron and a SC neuron to the optical stimulation in cdLSNr. Were the SNr neurons stimulated? Or the terminals from the CDt as described earlier in the manuscript.

Reply: We stimulated in cdLSNr the axon terminals of CDt neurons as described earlier in the manuscript. We have added some descriptions to clarify this as below (lines 201-203, p. 10).

“We therefore placed the recording electrode in the intermediate layer of SC when the optical stimulation activated the axon terminals of CDt neurons (which are all inhibitory neurons) on cdLSNr neurons.”

9. It can be read in the discussion: Accordingly, the information concerning a good object presented at a given position might be conveyed topographically along the CDt-cdLSNr-SC network in which individual neurons have similar receptive fields. Did the authors mapped the receptive fields in the CD, SNr and SC? It was described earlier (Pouderoux and Fretton 1979 Neurosci Lett; Nagy et al, 2003, 2006, Eur J Neurosci) that in the SNr and the CN of the feline brain large receptive fields can be found, which covers practically the whole visual field of the investigatd eye. What was the case in the monkey?

On the other hand, the receptive fields in the SCi are much smaller. How could the authors make the overlap between the receptive fields?

Reply: Previous studies showed the restricted visual field in CDt (Yamamoto et al., J. Neurosci., 2012), SNr (Hikosaka and Wurtz, J. Neurophysiol., 1983), and SC (Robinson, Vision Research, 1972) in primates. As the reviewer suspected, the receptive field of neurons in the CDt-related circuit, including SNr neurons, tends to be larger than that of SC neurons. But, the center of receptive field is correlated in SNr and SC neurons when the SNr-SC projection is present. Thus, we have revised the corresponding part in Discussion, because the previous version was unclear (lines 312-314, p. 16).

“It has been found that an SNr neuron projects to a specific site of SC where neurons have visual or oculomotor fields similar to the SNr neuron.”

In this study, we first checked the receptive field center of CDt, SNr, and SCi neurons by changing the object location at many times. First, we recorded an SNr neuron that

responded to optical stimulation, and checked the center of its receptive field. Based on this receptive field center, we recorded part of SC that is likely to have a similar receptive field based on the detailed spatial map of SC (Robinson, Vision Research, 1972). It was then likely that SNr and SC neurons with similar receptive fields were recorded simultaneously. We have added this procedural information in “Single-unit recordings” of Methods as below (lines 431-433, p. 22).

“For simultaneous recordings in both cdLSNr (by optrode) and SC (by recording electrode), we determined the SC recording site based on the receptive field map of SC after checking the receptive field of cdLSNr neurons.”

Minor points:

1. During the categorization of the CDt neurons did the authors checked the shape of the autocorrelograms? This is an important aspect of the categorization in the CD, too

Reply: Thank you for the suggestion. As reviewer suggested, we analyzed the shape of the autocorrelograms for categorization of CDt neurons, and re-analyzed the data based on this autocorrelograms. The autocorrelograms (as shown in Supplementary Fig. 1c-e) were similar to the results of a previous report (Adler et al., *Frontiers in Systems Neuroscience*, 2013). This re-analysis classified 7 MSNs, 6 TANs, 5 FSIs, and 3 unclassified neurons. We have revised Figure 1c and Supplementary Figure 1 according to this re-analysis. In addition, we have revised the corresponding part of Results as below (lines 86-88, p. 5).

“We identified three groups of CDt neurons based on their spontaneous firing rates, spike waveforms and autocorrelograms according to previous studies (Supplementary Fig. 1a-e)”

2. Line 193: which intermediate SC layer?

Reply: Though the intermediate layer is anatomically divided into SGI and SAI, we did not histologically examine the recording site within this SC layer. We defined this layer just as the intermediate layer by its electrophysiological property, because the recorded neurons showed both visual- and oculomotor-related activities. Further investigations are needed for more detailed sublayer-level analyses in the primate SC.

3. 247-248: after optical stimulation... What was stimulated here?

Reply: We stimulated in cdLSNr the axon terminals of CDt neurons. We have added a brief statement to clarify this as below (lines 257-259, p. 13).

“After optical stimulation in cdLSNr of the axon terminals of CDt neurons (Fig. 6a), the

SC neuron increased its firing rate (Fig. 6b, bottom), often with a burst of spikes (Fig. 6b, top),”

Reviewer #3 (Remarks to the Author):

Amita and colleagues used pathway specific optogenetics to determine the contribution the direct and indirect pathways from the caudate tail played in generating saccadic eye movements. In line with previous results using orthodromic stimulation, optogenetic stimulation of the CDt causes an inhibition of neurons in the SNr and GPe and preferentially causes inhibition is stable value coding neurons in these nuclei. Furthermore, optogenetic stimulation of the CDt terminals in the SNr caused an increase in firing in the superior colliculus and a facilitation of saccadic eye movements. Interestingly there was no effect of CDt terminal stimulation in the GPe on either SC neuronal activity or in the direction of saccadic eye movements.

Overall the results are well described and the data adds nicely to the previous work from this lab about stable value coding in the CDt pathway.

We just have a few minor points that we would like more clarification on. These minor points do not largely affect the quality of the study but should be clarified.

- The authors only recorded 7 SC units while stimulating CDt terminals in the GPe. None of these neurons were responsive to optogenetic stimulation of CDt fibers in the GPe. The authors did not report whether optogenetic stimulation of the CDt fibers in the GPe lead to a change in firing in the SNr. If this optogenetic stimulation did not result in a change in firing in the SNr then there would be no expected change in the activity of the SC units. Have the authors recorded in the SNr while stimulating the CDt fibers in the GPe? If they have or have not can they comment because this will affect the interpretation of the negative results obtained with stimulating the CDt fibers in the GPe. Essentially are these results a technical failure in that the SNr firing is not affected or is the SNr firing affected and the model for how this will impact the SC wrong?

Reply: This is a very important question which we need to solve in a future experiment. In the present work, we did not record neuronal responses from SNr while optically stimulating the CDt fibers in cvGPe, mainly because the optical stimulation did not cause a significant change in saccades, as shown in Supplementary Figure 8. This may be ascribed to the complex pattern of GPe neuron connectivity which has been shown in recent studies. Thus, we have added some descriptions in Discussion as below (lines 332-337, p. 17).

“This might be ascribed to the complex pattern of heterogeneous GPe neuron connectivity. In addition to the indirect pathways described above (CDt-cvGPe-(STN)-cdlSNr pathways), cvGPe neurons may have reciprocal connections

with STN neurons or striatal neurons, or have mutual inhibitory connections within GPe. Thus, the optical stimulation might reduce the presumed effect on the indirect pathways.”

- In both the GPe and the SNr there were both positive and negative value neurons that were sensitive to optogenetic stimulation. Can these be separated with examples of each shown in the supplemental information. This could be important as the population response presented in 3e does not appear to be significantly inhibited by either good or bad objects. Presumably this is due to a small number of positive coding neurons influencing the population response. If it is not due to the positive coding neuron and the negative coding population does not on average look like the example in 3c this needs to be discussed as it would not, then reflect the model they have previously and currently propose in 7. It would help in this regard if there was an analysis to show what proportion of neurons were significantly inhibited (below baseline) during the statistical test window.

Reply: Thank you for the important suggestion. As the reviewer suggested, we separately analyzed the GPe and SNr activities based on positive or negative value-coding neurons, and added the results as supplemental information (Supplementary Fig. 4). Our analysis revealed that the negative value-coding SNr neurons ($n = 28 / 66$) showed significant inhibitory responses to good objects (Supplementary Fig. 4a). The other neurons ($n = 38 / 66$), which showed excitatory responses to good and bad objects (Supplementary Fig. 4b), influenced the population response in Figure 3e as the reviewer pointed out. On the other hand, the positive value-coding GPe neurons ($n = 22 / 61$) showed significant inhibitory responses to bad objects but not to good objects (Supplementary Fig. 4c). The other neurons ($n = 39 / 61$) showed excitatory responses to good and bad objects (Supplementary Fig. 4d). These findings strongly support our hypothesis that the inhibitory response to good objects in SNr is caused by direct inhibition of CDt neurons, as depicted in Figure 7. We have added these data in Results as below.

> Results (lines 135-141, p. 7-8)

“These negative value-coding neurons, as a population, displayed significant inhibitory responses to good objects ($P^* = 0.026$, Wilcoxon signed-rank test) and significant excitatory responses to bad objects ($P^{***} < 0.001$, Wilcoxon signed-rank test) (Supplementary Fig. 4a), as previously demonstrated. In contrast, the other neurons (38 among 66) showed excitatory responses to both good and bad objects with short

latencies (Supplementary Fig. 4b), suggesting that they were likely to receive input not only from CDt, but also from other regions.”

> Results (lines 181-186, p. 9-10)

“These positive value-coding neurons displayed significant inhibitory responses to bad objects ($P^{**} = 0.0024$, Wilcoxon signed-rank test), but no significant responses to good objects ($P = 0.41$, Wilcoxon signed-rank test) (Supplementary Fig. 4c), as previously reported. By contrast, the other neurons (39 among 61) exhibited excitatory responses to both good and bad objects with short latencies (Supplementary Fig. 4d), suggesting that they were likely to receive input not only from CDt, but also from other regions.”

Reviewers' Comments:

Reviewer #1:

Remarks to the Author:

The authors adequately have addressed all of my concerns. I have no further comments or suggestions on this manuscript, except for a few remaining details:

p. 5, line 90. The authors' detail level can be more specific by indicating that caudate fast-spiking interneurons are thought to be parvalbumin-expressing GABAergic interneurons, not just GABAergic

References n°41. The first author's name properly spelled is "Saint-Cyr J.A."

Paul Apicella

Reviewer #2:

Remarks to the Author:

The authors addressed all my comments and concerns. The manuscript is now suitable for publication.

Reviewer #3:

Remarks to the Author:

The authors have satisfied all of my concerns and I now recommend the paper for publication.

REVIEWERS' COMMENTS:

Reviewer #1 (Remarks to the Author):

The authors adequately have addressed all of my concerns. I have no further comments or suggestions on this manuscript, except for a few remaining details:

p. 5, line 90. The authors' detail level can be more specific by indicating that caudate fast-spiking interneurons are thought to be parvalbumin-expressing GABAergic interneurons, not just GABAergic

Reply: Thank you, we have added the further information about the fast-spiking interneurons as below (p. 5, line 89-91).

“..., and fast-spiking interneurons (FSIs, presumed parvalbumin-expressing GABAergic interneurons, Supplementary Fig. 1e).”

References n°41. The first author's name properly spelled is "Saint-Cyr J.A."

Reply: Thank you, we corrected the first author's name as below.

“41. Saint-Cyr, J.A., Ungerleider, L. & Desimone, R. Organization of visual cortical inputs to the striatum and subsequent outputs to the pallidonigral complex in the monkey. Journal of Comparative Neurology 298, 129-156 (1990).”

Paul Apicella

Reply: We are grateful to you for your thoughtful and productive comments and suggestions. Your comments are very helpful for improving our manuscript.

Reviewer #2 (Remarks to the Author):

The authors addressed all my comments and concerns. The manuscript is now suitable for publication.

Reply: We are grateful to you for your thoughtful and productive comments and suggestions. Your comments are very helpful for improving our manuscript.

Reviewer #3 (Remarks to the Author):

The authors have satisfied all of my concerns and I now recommend the paper for publication.

Reply: We are grateful to you for your thoughtful and productive comments and suggestions. Your

comments are very helpful for improving our manuscript.